# Real-Time Scene-Adaptive Tone Mapping for High-Dynamic Range Object Detection

**Gongzhe Li**[1]    **Linwei Qiu**[2]    **Peibei Cao**[3]    **Fengying Xie**[2]    **Xiangyang Ji**[4]    **Qilin Sun**[1,5*]

[1] School of Data Science, The Chinese University of Hong Kong, Shenzhen, China
[2]Tianmushan Laboratory, Beihang University, Hangzhou, China
[3]School of Artificial Intelligence, Nanjing University of Information Science and Technology, China
[4]Department of Automation, Tsinghua University, Beijing, China
[5]Point Spread Technology, China
gongzheli1@link.cuhk.edu.cn,*sunqilin@cuhk.edu.cn

## Abstract

High-dynamic-range (HDR) images, with their rich tone and detail reproduction, hold significant potential to enhance computer vision systems, particularly in autonomous driving. However, most neural networks for embedded systems are trained on low-dynamic-range (LDR) inputs and suffer substantial performance degradation when handling high-bit-depth HDR images due to the challenges posed by extreme dynamic ranges. In this paper, we propose a novel tone mapping method that not only bridges the gap between HDR RAW inputs and the LDR sRGB requirements of detection networks but also achieves end-to-end optimization with downstream tasks. Instead of relying on the traditional image signal processing (ISP) pipeline, we introduce neural photometric calibration to regularize dynamic ranges and a scaling-invariant local tone mapping model to preserve image details. In addition, our architecture also supports performance transfer finetuning, enabling efficient adaptation from the LDR sRGB images to the HDR RAW images with minimal cost. The proposed method outperforms traditional tone mapping algorithms and advanced AI-ISP methods in challenging automotive HDR scenes. Moreover, our pipeline achieves real-time processing of 4K high-bit-depth HDR inputs on NVIDIA Jetson platforms.

## 1   Introduction

Real-world scenes can exhibit an immense dynamic range, reaching approximately 280 dB [1]. HDR cameras, by capturing a broader range of luminance, not only enrich visual content but also substantially enhance visual perception and decision-making—ultimately improving navigation safety [2]. State-of-the-art HDR sensors, such as the SONY IMX490 [3], deliver HDR RAW streams with an impressive dynamic range of up to 140 dB and preserve unparalleled levels of unprocessed detail. As a result, computer vision systems must handle challenging high-bit-depth HDR (e.g., 24-bit) scenes and rapidly changing lighting conditions in real-time, such as transitions when entering or exiting a tunnel.

Existing computer vision methods, particularly DNN-based object detection networks [4, 5, 6, 7, 8], are designed for low-dynamic-range (LDR) inputs processed through the traditional image signal processor (ISP). When applied directly to high-bit-depth HDR imagery, these systems suffer significant performance degradation 3.1 due to extreme luminance contrasts, leading to the collapse of feature extraction in neural networks. A common solution to this problem is to utilize professional HDR ISP, such as those described in [9, 10, 11], to convert HDR RAW images into LDR sRGB equivalents through a series of image processing steps. A critical component of these ISP is tone

39th Conference on Neural Information Processing Systems (NeurIPS 2025).

---

*Corresponding author.

mapping [12, 13, 14], which compresses the dynamic range while preserving as much detail as possible.

Handcrafted tone mapping approaches remain fundamentally optimized for human visual perception rather than machine vision tasks, often resulting in suboptimal performance when integrated with modern computer vision architectures. This mismatch also extends to traditional ISP pipelines, which inherit similar perceptual optimization constraints. Recent DNN-based AI-ISP methods [15, 16, 17, 18, 19, 20] largely follow traditional ISP pipelines for processing HDR RAW inputs. Some works [15, 16] implement neural approximations of key modules (e.g., AWB, Tone Mapping, CCM) for task-specific optimization, while others employ simplified tone curve parameterizations [21, 22] or proxy-based ISP tuning [18, 19, 17] guided by the downstream task. However, these methods remain constrained by ISP pipelines, which include redundant components and limit their adaptability to diverse HDR scenes in computer vision applications.

Modern edge computing platforms (e.g., NVIDIA Jetson) and neural network inference engines are exclusively optimized for LDR sRGB inputs. However, the extreme 24-bit dynamic range of HDR RAW images presents significant challenges for hardware-friendly operators, such as piecewise linear functions [23] or perceptual tone curves [13], which fail to represent the full dynamic range adequately. This introduces a gap between the HDR RAW inputs and these engines, necessitating innovative bridging solutions that can be effectively run on embedded systems.

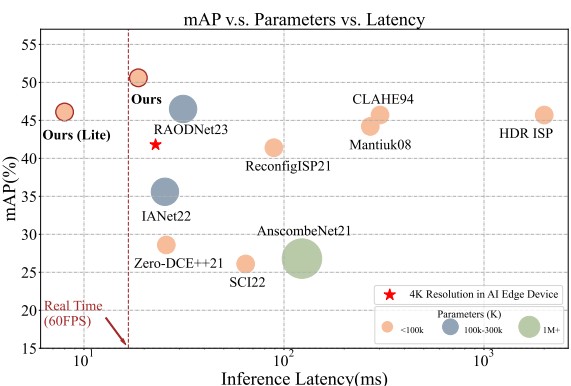

Figure 1: Comparison of detection performance and model complexity on the RoD dataset using Faster R-CNN (1280 × 1280 resolution). The symbol ★ indicates the performance of our Ours (Lite) model with 4K resolution input, achieving 45 FPS on NVIDIA Jetson platforms (16-bit float precision).

To bridge the gap between high-bit-depth HDR RAW inputs and LDR sRGB requirements of neural networks, we propose a lightweight and efficient tone mapping framework specifically optimized for HDR RAW object detection. The proposed architecture enables efficient computation and low latency, making it well-suited for edge platforms. To address changing lighting conditions in real-world environments, we propose neural HDR photometric calibration to dynamically regularize extreme dynamic ranges, normalize diverse radiance distributions into a unified scale, and enhance cross-scene generalization and detection performance. Our method outperforms both handcrafted tone mapping algorithms and modern AI-ISP baselines in autonomous driving datasets. Comprehensive ablation experiments demonstrate that our method significantly improves the detection performance on HDR RAW inputs. Furthermore, we show that our approach can process large-resolution HDR RAW data in real-time, achieving a balance between detection performance and inference latency. In summary, we make the following contributions:

- We propose a novel tone mapping framework for vision tasks to bridge the gap between high-bit-depth HDR RAW data and LDR sRGB input requirements for HDR object detection, specifically tailored for embedded platforms.

- We introduce a neural HDR photometric calibration method that dynamically regularizes extreme dynamic ranges through radiance-space unification.

- We design a lightweight local tone mapping model based on a scaling-invariant principle, enhancing detection performance while reducing runtime.

- We validate our method on an end-to-end object detection task, demonstrating its ability to efficiently process HDR RAW data while balancing detection performance and computational efficiency. Experimental results show that our method enables real-time processing of 4K HDR RAW videos.

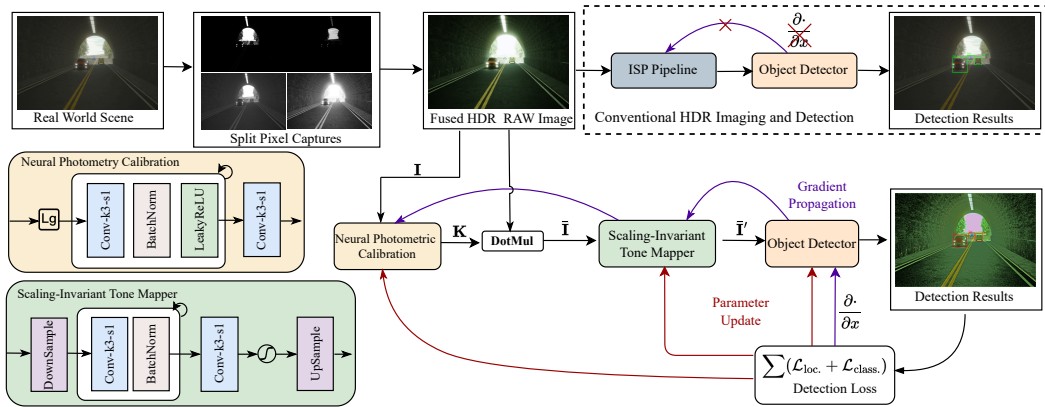

Figure 2: Conventional HDR RAW processing and detection are typically treated as separate tasks and optimized independently. We propose an alternative pipeline for HDR object detection, where tone mapping is optimized specifically for detection. The proposed pipeline regularizes the dynamic range and utilizes a scaling-invariant tone mapper to process both tasks simultaneously.

## 2 Related Works

### 2.1 High-Dynamic Range Imaging

HDR imaging techniques aim to capture a broader dynamic range of luminance. However, conventional CMOS sensors are limited in their luminance coverage, driving the development of HDR imaging solutions. Multi-exposure fusion (MEF) [24, 25] is a widely adopted approach in the industry, leveraging weight fusion strategies to recover the dynamic range from multiple exposures. While effective for photography, MEF methods are unsuitable for machine vision due to exposure latency and motion artifacts. For safety-critical applications like autonomous driving, researchers have explored single-shot HDR imaging solutions, including neural exposure control [26, 25], spatially varying pixel exposure [27, 28], and lighting modulation devices [29, 30], which extend dynamic range while avoiding motion artifacts.

While HDR imaging offers a broader dynamic range and richer tone reproduction, it necessitates novel methods to fully leverage its potential. Recently, [15] introduced a 24-bit HDR RAW image dataset captured using the IMX490 sensor, which employs advanced Split Pixel architecture [3] to capture an impressive dynamic range of 140 dB (approximately 24 stops). This extreme dynamic range significantly surpasses that of existing datasets, such as LoD [31] (14-bit), PASCAL RAW [32] (14-bit), and RhoVision [33] (12-bit), presenting a considerable challenge to current computer vision systems.

### 2.2 Post-Captured Tone Mapping Algorithms

Traditional computer vision networks are typically tailored for LDR images processed by the Camera ISP pipelines [10, 11, 34]. Tone mapping algorithms [12, 14, 13] are a critical component of camera ISP, compressing dynamic range and equalizing luminance to render HDR content into visually pleasing LDR images. However, these algorithms and ISP systems are primarily designed for human perception, often including unnecessary steps and being computationally expensive. While effective for visual aesthetics, they are suboptimal for machine vision tasks such as object detection. To address these limitations, DNN-based AI-ISP approaches [15, 16, 17] introduce learnable ISP pipelines that leverage neural networks for HDR tone mapping. These methods can be jointly optimized with downstream neural networks via gradient propagation, outperforming traditional ISP systems in vision tasks. Additionally, differentiable approximations [19, 18, 17] model hardware ISP using techniques such as neural architecture search [19] or parameter mapping [18] to simulate real ISP systems. Unsupervised low-light enhancement methods [21, 22] also act as independent tone mappers, equalizing luminance through tone curves. While these methods reduce computational costs, their reliance on specific models limits generalization under varying illumination conditions. In contrast, our method employs HDR photometric calibration and an efficient tone mapping network tailored

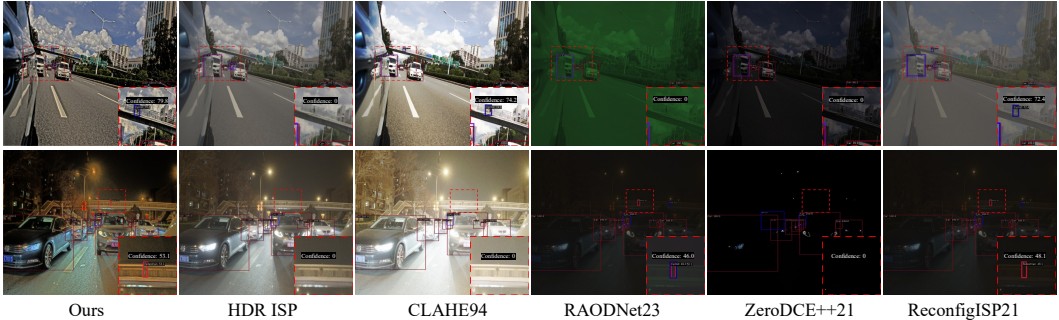

| Ours | HDR ISP | CLAHE94 | RAODNet23 | ZeroDCE++21 | ReconfigISP21 |

Figure 3: Visual comparison of different methods on HDR RAW images. The first row illustrates daytime scenes, while the second row showcases night scenes. Our method demonstrates superior performance compared to the other methods. Zoom in to see details.

for HDR perception, eliminating dependence on ISP meta-architectures and avoiding the need for complex design.

## 2.3 Perception Networks on Embedded Systems

Practical vision applications, such as those in automotive and robotics, are often deployed on embedded systems (e.g., NVIDIA Jetson, ARM Core) with limited computational resources, necessitating optimized neural network inference. Lightweight architectures [5, 35] are specifically tailored for AI edge platforms, enabling efficient inference under resource constraints. To further enhance performance, quantization techniques convert floating-point weights into low-precision integers [36, 37], while pruning methods [38, 39] eliminate redundant layers to reduce memory and computation overhead. These optimizations make neural networks more suitable for embedded systems, but often overlook the challenges posed by HDR RAW inputs.

Most deployed models are trained on LDR sRGB images (e.g., 8-bit), making them incompatible with HDR RAW data (e.g., 24-bit). This mismatch can result in ineffective feature extraction and degraded performance for HDR perception tasks [15, 40, 33]. To address this issue, we propose a novel tone mapping method that is jointly optimized with the detector, enabling efficient and accurate processing of HDR RAW data on embedded systems. Our method bridges the gap between HDR imaging and embedded perception, enabling resource-constrained platforms to handle HDR tasks effectively.

## 3 Method

In this section, we first analyze the impact of dynamic range on detection performance and propose a neural photometric calibration method to regularize it. Additionally, we present a scale-invariant tone mapping approach designed to optimize image details, guided by the requirements of the downstream detector. An overview of the proposed method is illustrated in Fig. 2.

### 3.1 Dynamic Range Regularization

**Dynamic Range and Detection Performance Analysis.** Although 24-bit HDR RAW images retain rich details that could theoretically enhance detection performance, experiments reveal a counterintuitive issue: directly applying detectors to HDR images causes gradient instability (see Table 1). We explore the relationship between image dynamic range and gradient propagation in neural networks. Specifically, we model pixel distributions of an HDR image using a Gaussian Mixture Model (GMM) [41] to approximate the histogram across the entire dynamic range. The distribution $p(x)$ of an image $x$ is explicitly expressed as a weighted mixture of $N$ Gaussian component densities $\mathcal{N}(\mu_i, \sigma_i)$, designed to approximate the pixel histogram of the HDR input:

$$p(x) \approx \sum_{i=1}^{N} \omega_i \cdot \mathcal{N}(\mu_i, \sigma_i) \tag{1}$$

where $\omega_i$ are the weights, and $\mu_i$ and $\sigma_i$ are the mean and standard deviation of the $i$-th Gaussian component. $N$ is the number of components. Next, we propose a minimal model consisting of a single-layer CNN with MSE loss to analyze the effects of dynamic range on gradient propagation dynamics. This simplified model allows for the theoretical derivation of the gradient relationship:

$$\frac{\partial \mathcal{L}}{\partial \omega} = \frac{\partial}{\partial \omega}(\omega * x + c - \hat{y})^2 \approx \mathbb{E}(x) + \mathbb{E}(x^2) \approx \sum_i^N \omega_i(\mu_i^2 + \sigma_i^2 + \mu_i) \quad \# \quad d_i = \frac{\mu_i + \sigma_i}{\mu_i - \sigma_i}$$

$$\approx \sum_{i=1}^N \pi_i \left[ \sigma_i^2 + \left( \sigma_i \cdot \frac{d_i + 1}{d_i - 1} \right)^2 + \sigma_i \cdot \frac{d_i + 1}{d_i - 1} \right] \propto \sum_{i=1}^N \pi_i \left( \frac{d_i + 1}{d_i - 1} \right)^2 \tag{2}$$

Here, the single-layer CNN is expressed as $\omega * x + c$, where $\omega$ represents kernel weights, $x$ is the input, $c$ is bias, $*$ denotes the convolution operator. $\mathcal{L}$ is the MSE loss and $\hat{y}$ denotes the ground truth label. $d_i$ represents the ratio of the maximum value to the minimum value in a single Gaussian component, approximating the dynamic range. As shown in Eq. (2), the gradient is proportional to the square of the dynamic range, indicating that HDR images lead to higher gradient fluctuations compared to LDR images. This fluctuation makes it challenging for neural networks to extract effective features, leading to instability in gradient propagation [15, 42, 33]. Detailed proofs are provided in the supplemental material.

**HDR Photometric Calibration.** The gradient fluctuations in HDR images motivate us to regularize their dynamic range to stabilize training. HDR photometric calibration is commonly used to normalize pixel values to the irradiance of the real-world scene, projecting HDR images into a unified radiance space while preserving fine details. Previous studies [43, 44, 45] have shown that HDR photometric calibration enhances detail reproduction and improves visual perception. The HDR photometric calibration process can be expressed as:

$$\bar{\mathbf{I}} = V \cdot \mathbf{R}, \tag{3}$$

where $\bar{\mathbf{I}}$ represents the calibrated image, $\mathbf{R}$ denotes the scene radiance, and $V$ models the optical effects. This calibration is typically modeled as a linear transformation.

## 3.2 Neural Photometric Calibration

In practice, measuring real photometry for HDR image calibration is challenging. To address this issue and adapt to different lighting conditions, we need to make educated guesses about the minimum and maximum radiance values in the original scene. Hence, we propose *Neural Photometric Calibration*, a novel method designed to approximate Eq. (3) and estimate its adaptive parameters through end-to-end optimization. Specifically, we define a learnable transformation function as:

$$\bar{\mathbf{I}} = (\mathbf{K} - b) \cdot \mathbf{R} + b \tag{4}$$

where $\bar{\mathbf{I}}$ is calibrated HDR RAW image, $\mathbf{R}$ is captured scene radiance. $\mathbf{K}$ is the scale map, and $b$ is bias term. To handle complex scene illumination, we introduce a linear interpolation-based modification to adaptively predict the scale map:

$$\mathbf{K} = \boldsymbol{\alpha} \cdot R_{\text{Day}} + (\mathbf{E} - \boldsymbol{\alpha}) \cdot R_{\text{Night}} \tag{5}$$

Here, $\mathbf{K}$ is controlled by scene key radiance [13] for both day and night scenes (e.g. $R_{\text{Day}}$ and $R_{\text{Night}}$), ensuring compatibility across different lighting scenes. $\mathbf{E}$ is identity matrix, and $\boldsymbol{\alpha}$ is a weight map constrained by a sigmoid function:

$$\boldsymbol{\alpha} = \text{sigmoid}\left( \text{FE}\left( \mathbf{R}_\downarrow \right) \right) \tag{6}$$

where the uncalibrated image $\mathbf{R}$ is first downsampled ($\downarrow$) to low resolution (LR) and then passed through a feature extractor FE to generate the weight map $\boldsymbol{\alpha}$.

The radiance in sunlight can be as much as a million times more intense than at night ($R_{\text{Day}} \gg R_{\text{Night}}$). This numerical disparity introduces instability when directly predicting $\mathbf{K}$ using the convolution layer in Eq. (5). Thus, we enforce the convolution layer to predict log scale $\mathbf{S} := \log \mathbf{K}$:

$$\mathbf{K} = \{10^{\mathbf{s}}\}_\uparrow, \quad \mathbf{S} = \boldsymbol{\alpha} \cdot \log R_{\text{Day}} + (\mathbf{E} - \boldsymbol{\alpha}) \cdot \log R_{\text{Night}} \tag{7}$$

The logarithmic mapping allows us to regress compact values, where $10^{\mathbf{S}}$ is resolved to the radiance scale, then the scale map is upsampled ($\uparrow$) to the original size. In this paper, we set $R_{\text{Day}}$ and $R_{\text{Night}}$ approximately at $10^7 \text{cd/m}^2$ and $10^4 \text{cd/m}^2$, respectively, as cited from [1], and the bias term $b = 10$ in our experiments.

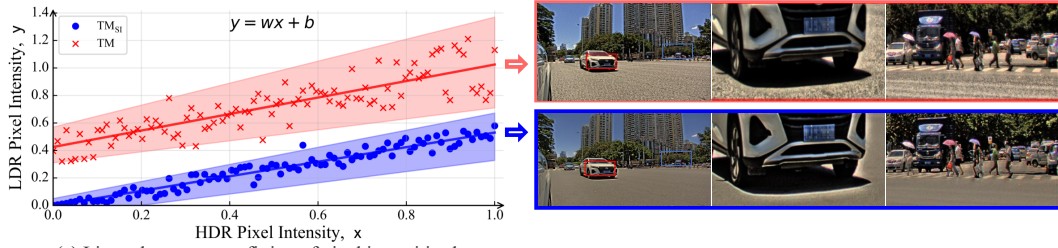

(a) Linear least squares fitting of pixel intensities between LDR and HDR images.

(b) Visual Comparison between $\text{TM}_{\text{SI}}$ and TM.

Figure 4: (a) Linear least squares fitting on pixel intensity from processed LDR image and input HDR images. TM exhibits a large bias gain, while $\text{TM}_{\text{SI}}$ only introduces a scale gain. (b) The comparison of the LDR images shows that $\text{TM}_{\text{SI}}$ produces smooth regions and effectively enhances object visibility, while TM often results in background noise.

## 3.3 Scaling-Invariant Tone Mapper

We propose a lightweight and effective local tone mapper that leverages the *scale-invariance property* of neural networks. We first prove that a bias-free neural network [46, 47] (*i.e.* , no additive bias terms) with $L$ layers, composed of convolution weights $\{K\}_i^L$, ReLU activation, and Batch Normalization, is scaling-invariant:

$$
\begin{aligned}
f_{\text{SI}}(\alpha x) &= \text{RELU} \circ \text{BN}_L \circ K_L * \cdots \circ \text{RELU} \circ \text{BN}_1 \circ K_1 * (\alpha x) \\
&= \text{RELU} \circ \text{BN}_L \circ K_L * \cdots \circ \text{RELU} \left( \alpha \cdot \gamma_1 \cdot \frac{\cdot K_1 * x - \mu_1}{\sigma_1} \right) \\
&= \text{RELU} \circ \text{BN}_L \circ K_L * \cdots \circ \alpha \cdot \text{RELU} \left( \gamma_1 \cdot \frac{K_1 * x - \mu_1}{\sigma_1} \right) \\
&= \alpha \cdot \text{RELU} \circ \text{BN}_L \circ K_L * \cdots \circ \text{RELU} \circ \text{BN}_1 \circ K_1 * x \\
&= \alpha \cdot f_{\text{SI}}(x).
\end{aligned}
\tag{8}
$$

where $\circ$ denotes the cascading of network layers, $*$ is a convolution operator, and BN denotes batch normalization, defined as $\text{BN(x)} = \gamma \cdot (x - \mu)/\sigma$, where $\gamma, \mu, \sigma$ are statistic parameters. Since convolution layers rely on local dependencies, this property inherently preserves local relationships. This design ensures that the scaling-invariant network functions as a local tone mapper without requiring additional modifications. Scale-invariance is a critical property for tone mapping, as it ensures consistent reproduction across varying dynamic ranges.

The input HDR RAW image is processed using a scale-invariant tone mapper ($\text{TM}_{\text{SI}}$), and the resulting LDR image is fed into the object detector (OD) to obtain object information. This operation is formally expressed as:

$$
(d, c, s) = \text{OD}(\text{TM}_{\text{SI}}(\bar{\mathbf{I}}))
\tag{9}
$$

where the $d$ are the detected bounding boxes and $c$ and $s$ the corresponding inferred classes and confidence scores. And in the following text, we will refer to the scaling-invariant tone mapper as $\text{TM}_{\text{SI}}$ and the scaling-variant tone mapper as TM.

**Architecture Details.** The detailed implementation is illustrated in Fig. 2. The neural photometric calibration module consists of 3-layers of *Conv-ReLU* followed by a sigmoid activation. The tone mapping model $\text{TM}_{\text{SI}}$ consists of 4 layers of *Conv-BN-ReLU*, each with 32 channels. To preserve the scaling-invariant property, all bias terms are removed from the network. Additionally, we introduce a lightweight variant with 16 channels, referred to as Ours (Lite), to enable efficient inference for large-resolution inputs on edge devices.

## 3.4 End-to-End Optimization

**Tone Mapper Pretraining**. In our method, the input HDR RAW images are processed to compress the dynamic range before passing them to the detector, making effective weight initialization critical for end-to-end optimization. To pretrain the tone mapper, we use the Normalized Laplacian Pyramid Distance (NLPD), a self-supervised loss function commonly applied in perceptual image quality optimization [53, 43]. The NLPD loss compresses the dynamic range while preserving tone details

Table 1: Quantitative comparison of basic detectors (Faster R-CNN and YOLOv3) on the ROD dataset, evaluated using mAP, mAR, AP50, and AP75 metrics. The best results are highlighted in **bold**, while the second-best results are indicated with an underline. **NAN** indicates that the results do not converge.

| Methods | Method Group | Faster R-CNN [4] | | | | YOLOv3 [5] | | | | Latency(ms) | FLOPs(G) |
|---|---|---|---|---|---|---|---|---|---|---|---|
| | | mAP | AP50 | AP75 | mAR | mAP | AP50 | AP75 | mAR | | |
| HDR RAW | Direct Input | —NAN— | | | | —NAN— | | | | - | - |
| HDR ISP [9] | Handcrafted ISP | 45.7 | 69.5 | 50.1 | 54.9 | 42.2 | 68.1 | 45.6 | 50.5 | - | - |
| Mantiuk08 [12] | Handcrafted Tone Mapping | 45.6 | 69.1 | 49.7 | 53.3 | 44.8 | 71.0 | 49.1 | 52.8 | - | - |
| CLAHE94 [14] | | 44.2 | 68.2 | 49.1 | 53.0 | 43.0 | 69.2 | 46.5 | 51.4 | - | - |
| ReconfigISP21 [19] | End-to-End ISP | 41.4 | 63.2 | 45.6 | 49.7 | 40.9 | 60.0 | 44.0 | 48.4 | 92.7 | 455.14 |
| Zero-DCE++21 [21] | Low-Light Enhance | 28.6 | 47.4 | 30.2 | 38.5 | 27.5 | 49.0 | 27.5 | 36.3 | 29.8 | 16.51 |
| SCI22 [22] | | 26.1 | 43.7 | 27.0 | 35.6 | 25.1 | 45.6 | 25.1 | 33.5 | 64.3 | 283.14 |
| AnscombeNet21 [18] | AI-ISPNet | 26.8 | 43.6 | 27.5 | 37.8 | 22.0 | 41.5 | 23.1 | 30.4 | 123.3 | 382.04 |
| IANet22 [16] | | 37.6 | 59.5 | 40.1 | 47.0 | 33.1 | 56.4 | 34.4 | 41.9 | 31.3 | 0.96 |
| RAODNet23 [15] | | 46.1 | 69.4 | 50.6 | 55.4 | 42.9 | 65.8 | 46.5 | 50.9 | 41.2 | 0.81 |
| RawOrCooked23 [42] | | 34.8 | 55.3 | 34.4 | 44.4 | 28.2 | 46.3 | 28.4 | 32.9 | 55.1 | 12.89 |
| Ours (Lite) | End-to-End Tone Mapping | 47.8 | 71.7 | 52.9 | 56.9 | 42.6 | 68.5 | 46.6 | 51.3 | 8.8 | 23.40 |
| Ours | | 49.8 | 73.3 | 55.6 | 58.7 | 45.1 | 71.4 | 50.2 | 53.0 | 25.4 | 62.11 |

Table 2: Performance comparison using per-formance transfer strategy on RoD dataset.

| Pretrained Weights | Methods | mAP | mAR |
|---|---|---|---|
| COCO[48] | RAWAdapter24 [49] | 12.6 | 25.5 |
| | AdaptiveISP24 [17] | 22.5 | 33.5 |
| | Ours | **30.6** | **47.5** |
| Object365[50] | RAWAdapter24 [49] | 15.7 | 28.9 |
| | AdaptiveISP24 [17] | 24.7 | 36.1 |
| | Ours | **38.8** | **50.7** |

Table 3: Impact of image texture metric and detection results.

| Methods | Image [51] Contrast($\uparrow$) | Entropy [52] of GLCM($\uparrow$) | mAP |
|---|---|---|---|
| HDR RAW | 0.00 | 0.00 | -NAN- |
| SCI22 [22] | 0.18 | 19.17 | 26.1 |
| Zero-DCE++21 [21] | 0.18 | 20.07 | 28.6 |
| IANet22 [16] | 0.19 | 21.92 | 37.6 |
| RAODNet23 [15] | 0.22 | 24.32 | 46.1 |
| Ours | **0.41** | **24.63** | **49.8** |

and effectively enhances local contrast, which we observed to be beneficial for improving object detection performance.

**Performance Transfer Strategy**. Our method can also act as a domain adapter [49, 17] to transfer performance from the LDR sRGB domain to the HDR RAW domain. Through the performance transfer strategy, we can achieve high performance with only a few training sources by utilizing pretrained weights on LDR sRGB. In this setting, we optimize only the tone mapper and the detection head, while keeping the remaining components frozen.

**Loss Function.** In our experiments, training is guided by detection losses commonly used in object detection pipelines [4, 5]:

$$\mathcal{L}_{\text{total}} = \mathcal{L}_{\text{obj.}} + \mathcal{L}_{\text{class.}} \tag{10}$$

Here, the total loss $\mathcal{L}_{\text{total}}$ consists of a location regression loss ($\mathcal{L}_{\text{obj.}}$) and a classification loss ($\mathcal{L}_{\text{class.}}$), both of which are widely applicable in detection tasks.

# 4 Experiments

## 4.1 Experimental Setups

**Dataset.** We evaluate our method on the RoD dataset [15], which contains 20,089 24-bit HDR RAW images. Unlike RAODNet [15], we test our proposed method on mixed scenes to validate its effectiveness and generality. Note that the published dataset is more challenging than in the paper.

**Implementation Details.** We employ two widely used object detectors: Faster R-CNN (ResNet50) [4] and YOLOv3 (DarkNet53) [5]. Our implementation is based on the *MMDetection* [54] codebase. Following the setup in [15], HDR RAW images are processed with linear demosaicing [55, 56] to restore color channels and resized to $1280 \times 1280$ size. During training, we apply random flipping for data augmentation, use a batch size of 8, and train for 14 epochs with an initial learning rate of 1e-2. The learning rate is decayed by a factor of 10 at epochs 8 and 11. And we use Faster R-CNN [4] as the main detector for the following ablation studies. To accelerate adaptation to the HDR RAW domain, we initialize the model with COCO [48] pretrained weights.

**Evaluation.** We evaluate performance using mean Average Precision (mAP) and mean Average

Table 4: Performance comparison under different scenes of Faster R-CNN.

| Scenes | HDR ISP [10] | IANet21 [16] | RAODNet23 [15] | Neural Calibration w/o | Neural Calibration w |
|---|---|---|---|---|---|
| Day | 32.0 | 43.4 | 35.9 | 36.7 | **40.3** |
| Night | 37.9 | 31.6 | 45.6 | 38.5 | **45.8** |
| Mixed Scene | 45.7 | 37.6 | 46.1 | 40.7 | **49.8** |

Table 5: Performance comparison using performance transfer strategy on RoD dataset.

| Scaling Invariant | RoD[15] | | | RoD $\rightarrow$ RhoVision [33] | | |
|---|---|---|---|---|---|---|
| | mAP | AP75 | AP50 | mAP | AP75 | AP50 |
| ✘ | 49.0 | 72.5 | 52.4 | 14.3 | 24.5 | 13.5 |
| ✔ | **49.8** | **73.3** | **55.6** | **26.5** | **55.9** | **27.9** |

Table 6: Impact of texture metric and detection results on RoD dataset.

| Detectors | Methods | mAP | AP50 | AP75 | mAR |
|---|---|---|---|---|---|
| Sparse RCNN [7] | HDR ISP [10] | 45.0 | 68.7 | 49.0 | 61.0 |
| | Ours | **51.3** (+6.3) | **74.7** | **57.1** | **66.4** |
| Deformable DETR[8] | HDR ISP [10] | 50.2 | 73.6 | 57.5 | 64.7 |
| | Ours | **55.0** (+4.8) | **78.4** | **63.4** | **68.2** |

Recall (mAR) across all Intersection over Union (IoU) thresholds, along with Average Precision (AP) at IoU thresholds of 0.5 (AP50) and 0.75 (AP75). Additionally, we test model complexity in terms of the parameters (K), computational complexity (FLOPs), and inference latency (ms) on NVIDIA Jetson platforms.

## 4.2 Comparison Experiments

**Quantitative Comparison.** We compare the proposed method with state-of-the-art (SOTA) methods for HDR object detection; the quantitative results are presented in Table 1. We categorize these comparison methods into five groups: Handcrafted ISP [10], Handcrafted Tone Mapping [12, 14], End-to-End ISP [19], Low-Light Enhancement [21, 22], AI-ISPNet [18, 16, 15, 42], and End-to-End Tone Mapping (our methods). The HDR ISP pipeline is adapted from the post-processing of HDRPlus [9, 10], a professional ISP pipeline designed for HDR image rendering. Directly using HDR RAW images as input for object detectors results in **NaN outcomes** due to their extreme dynamic range, where most pixels tend toward small values. This prevents the neural network from extracting sufficient features, highlighting the current models' inability to effectively capture meaningful information. In contrast, tone mapping algorithms such as Mantiuk08 [12] and CLAHE94 [14] compress the dynamic range while preserving image details, resulting in performance that is comparable to SOTA and significantly benefits the detection task. Differentiable ISP methods, such as AnscombeNet21 [18], model ISP transformations in latent space but lack specific tone mapping operations, resulting in lower performance. ReconfigISP21 [19] employs proxy optimization to fine-tune a simple ISP system, achieving results comparable to HDR ISP pipelines. Unsupervised low-light enhancement [21, 22] achieves lower performance because it cannot effectively handle high dynamic range scenes, and the processed image is still dark. As for AI-ISP methods [16, 15, 42], IANet22 [16] employs a piece-wise linear curve as a tone mapper, which is only suitable for monotone lighting scenes, resulting in low performance. RAODNet23 [15] uses both global and local tone curves to compress the dynamic range, achieving results comparable to state-of-the-art methods. RAWorCooked23 [42] introduces a learnable contrast correction function, but it struggles with handling extreme dynamic ranges. Our method outperforms all comparison methods, achieving improvements of 3.7% in Faster R-CNN and 1.6% in YOLOv3 compared to the second-best performance, respectively. Additionally, it shows a 4.1% improvement compared to the HDR ISP pipeline. These comparison results demonstrate that our proposed method effectively explores potential information in HDR RAW images, leading to improved detection performance.

**Qualitative Comparison.** In Fig. 3, we compare our method with the baseline by visualizing detection results across both day and night scenes in the RoD dataset. For daytime scenes, our method enhances regional dynamics and highlights potential objects, making detection easier. In night scenes, it adjusts the contrast across regions, making candidate areas more prominent. Our method can handle varying lighting conditions, thanks to neural photometric calibration. This process normalizes the radiance space and optimizes tone mapping for the downstream detector, offering an optimal solution for machine vision perception. More visual comparisons are provided in the supplemental material.

**Inference Latency Comparison.** We evaluate the parameters and inference latency of comparison methods in Fig. 1. Our method utilizes a plain feedforward architecture, where each layer takes the output of its preceding layer as input and passes its output to the following layer. This design ensures a favorable accuracy-speed trade-off, even though it does not use the fewest number of parameters.

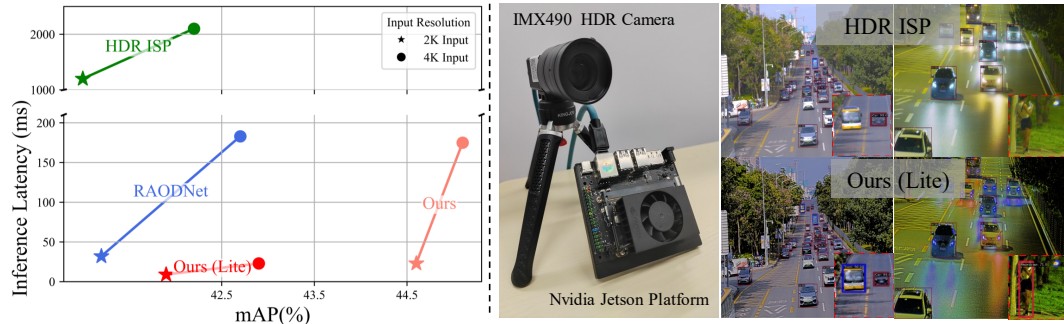

Figure 5: Left: Performance comparison on NVIDIA Jetson platforms. Right: Illustration of our hardware prototype, integrating HDR cameras and the NVIDIA Jetson platform for real-world evaluation.

Although methods such as [16, 15, 22] reduce FLOPs by downsampling the input resolution, they also lead to a loss of image details, which can potentially harm performance. In comparison, our approach strikes the optimal balance between efficiency and performance.

### 4.3 Ablation Study and Discussion.

**Ablations on Performance Transfer Strategy.** We evaluate the performance transfer strategy as described in Sec. 3.4. Specifically, we initialize the object detector with publicly pretrained weights [48, 50] and fine-tune only the *tone mapper* and *detection head* for 5 epochs. We also evaluate the RAW domain adapter methods [49, 17], with the results shown in Table 2. Our approach optimizes only 33.6% of the total parameters of Faster R-CNN (tone mapper: 0.10% + detection head: 33.6%), yet achieves state-of-the-art performance compared to other methods. Our method stands out with its simple yet effective architecture, avoiding ad-hoc designs while achieving strong performance.

**Ablations on Scene Generalization.** To evaluate cross-scene generalization, we divided the RoD dataset [15] into two subsets based on scene lighting. HDR ISP [9] is introduced as a baseline due to its ability to generate consistent LDR images, and we also introduce AI-ISP methods [16, 15]. The results, shown in Table 4, indicate that IANet22 [16] and RAODNet23 [15] outperform HDR ISP in a single scene. However, neither method matches HDR ISP performance across other test scenes. These limitations arise because they rely on the PWL curve as a tone mapping module, which is only effective for monotonous lighting scenes. We also conduct an ablation study on neural photometric calibration. This module improves performance across all test scenes, demonstrating its effectiveness in radiance normalization.

**Ablations on Advanced Detector.** We also evaluate advanced detectors (Sparse-RCNN [7] and Deformable DETR [8]) in our method. The experimental results in Table 6 show that our method significantly improves detection performance on HDR RAW images, outperforming HDR ISP by 6.3% and 4.8%, respectively. These results demonstrate that our method exhibits improved generalization across various downstream detectors.

**Analysis of Scaling-Invariant Tone Mapper.** To evaluate the generalization ability of our proposed scaling-invariant tone mapper, we train the model on the RoD dataset [15] and test it on the RhoVision dataset [33], which contains HDR RAW images captured by the same sensor in different scenes. The results in Table 5 show that the scaling-invariant tone mapper achieves superior generalization performance on the test dataset. Additionally, we evaluate the relationship between pixel intensities in the processed LDR image and the input HDR image using linear least squares fitting. As shown in Fig. 4 (left), the TM introduces a large bias gain, with the magnitude being much larger than that of the scale gain. This generates more noise and reduces the contrast between objects and the background, which may negatively affect generalizability.

From another perspective, tone mapping is necessary to map the HDR image onto the LDR image space. This space must encompass all possible rescalings of those images, including the origin. This implies that $\text{TM}(\alpha \cdot x) = \alpha \cdot \text{TM}(x), \forall x > 0$. To achieve this, we design a bias-free CNN, which ensures that any negative entries in the input are set to zero. Multiplying by a nonnegative constant does not change the sign of the entries in a vector. Therefore, the bias-free network remains scale-invariant and can rescale the image without introducing any bias.

**Analysis of Image Texture.** We evaluate image texture metrics using image contrast [51] and the entropy of the Gray-Level Co-occurrence Matrix (GLCM) [52]. As shown in Table 3, our method enhances image contrast and highlights object regions, leading to improved detection performance. These results underscore the importance of texture information for detection tasks.

## 4.4 Real World Evaluation.

We evaluate the latency and mAP performance of YOLOv3 [5] on the NVIDIA Jetson AGX Orin (16-bit float precision), with the results shown in Fig. 5 (left). Our method (Lite) achieves a runtime of 22 ms (45 FPS) for 4K resolution (4096 × 2160), striking an excellent balance between performance and real-time inference. Additionally, we collect an HDR RAW video dataset from real-world driving scenes to validate the proposed method. Fig. 5 (right) shows our hardware prototype for real-world evaluation, while the right side provides an example dataset used in this study.

# 5 Conclusion and Limitation

Real-world scenes present significant challenges for computer vision due to their wide dynamic range of luminance. Instead of relying on conventional ISP pipelines, we propose a real-time, scene-adaptive tone mapping method that optimizes image details for HDR detection. Our approach introduces neural photometric calibration to regularize the dynamic range, ensuring generalization across diverse lighting scenes. Additionally, a scaling-invariant tone mapping module is integrated into an end-to-end trainable vision pipeline, optimizing image details through joint training. With its efficient architecture, our method supports a low-cost performance transfer strategy, enabling adaptation from the LDR sRGB domain to the HDR RAW domain by finetuning only a few parameters. Experiments demonstrate that the proposed method outperforms SOTA methods in both detection performance and inference latency, while also achieving real-time processing of 4K HDR RAW inputs.

Although our proposed method effectively handling a variety of lighting conditions and significantly outperforming comparison methods, it still requires cascading with detectors for joint processing of HDR RAW input. We believe that a well-designed detector can directly handle HDR data by fine-tuning certain parameters to adapt to HDR RAW data without any additional processing, thus further exploiting the information from sensor data and improving both the efficiency and effectiveness of detection. This is the direction of our future work.

# 6 Acknowledgement

This work was supported in part by the National Nature Science Fundation of China (62375233,62302423), Shenzhen Pivot Funding (2024TC0036), and Science and Technology Program ZDSYS202211021111415025. We also express our sincere gratitude to Prof. Kede Ma of the City University of Hong Kong for his insightful discussions and valuable comments.

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

# Supplementary Material

In this file, we provide the following supplementary studies:

- Details of Implemented HDR ISP pipeline appendix A.
- The proof of approximation to Dynamic Range appendix B.
- The proof of scaling-invariant Tone Mapping appendix C.
- More Comparison Experiments.
- More Ablation Studies. appendix D.
- More Vision Comparison. appendix E.
- Real World Evaluation. appendix F.

## A   HDR ISP Details

In this section, we describe the HDR ISP pipeline used in the comparison method [9]. This pipeline consists of a series of operations, as illustrated in Fig. 6. We follow the implementation described in [9, 10], with modifications applied to the modules preceding the tone-mapping algorithms. The intermediate results of key components (indicated by the red dashed lines in Fig. 6) are shown in Fig. 7, which demonstrate how the HDR data is transformed into a visually appealing LDR image after undergoing several nonlinear operations. Next, we introduce the key steps in this process.

**Black-level Correction**: We should subtract an offset from all pixels so that pixels receiving no light have a value of zero. This offset is obtained from optically shielded pixels on the sensor.

$$I_{\text{blc}} = I - I_{\text{bl}} \tag{11}$$

where $I_{\text{bl}}$ is the black level.

**Anti-Aliasing filter**: An anti-aliasing filter is a type of low-pass filter that prevents aliasing components from being sampled.

$$I_{\text{aaf}} = I * k_{\mathbf{aaf}} \tag{12}$$

where $k_{\mathbf{aaf}}$ is $5 \times 5$ filter kernel, having non-zero elements only at the corners and center. The kernel is defined as: $k_{\mathbf{aaf}} = 1/16 \cdot \begin{bmatrix} 1 & ... & 1 \\ ... & 8 & ... \\ 1 & ... & 1 \end{bmatrix}$.

**Auto White Balance**: The AWB (Auto White Balance) module is responsible for adjusting the image to ensure that the four (RGGB) channels are linearly scaled, so that grays in the scene correspond to grays in the image. These scaling factors are calculated using the Gray-World white balance algorithm, which adjusts the pixel values based on the gray-world assumption. This assumption posits that the average of all color channels should produce a neutral gray image.

$$\begin{bmatrix} I_{R'} \\ I_{Gr'} \\ I_{Gb'} \\ I_{B'} \end{bmatrix} = \begin{bmatrix} g_R & 0 & 0 & 0 \\ 0 & g_{Gr} & 0 & 0 \\ 0 & 0 & g_{Gb} & 0 \\ 0 & 0 & 0 & g_B \end{bmatrix} \begin{bmatrix} I_R \\ I_{Gr} \\ I_{Gb} \\ I_B \end{bmatrix} \tag{13}$$

where the parameters $g_R, g_{Gr}, g_{Gb}, g_B$ are color gain, which can be derived by the gray world algorithm.

**Demosaic**: Demosaicing converts a Bayer raw image into a full-resolution linear RGB image, preserving texture details. We use a combination of techniques from the Malvar algorithm [55].

$$I_{\text{R,G,B}} = \text{Demosaic}(I_{(\text{R,Gr,Gb,B})}) \tag{14}$$

where $I_{(\text{R,Gr,Gb,B})}$ denote the single channel bayer array, $I_{\text{R,G,B}}$ denote the complete 3-channels RGB image.

**Local Tone Mapping**: The LTM (Local Tone-Mapping) block simulates the exposure fusion algorithm [10, 24] by brightening darker areas while ensuring that brighter content remains unsaturated.

$$I_{\text{ltm}} = \sum_{i}^{n} I_i \cdot w_i \tag{15}$$

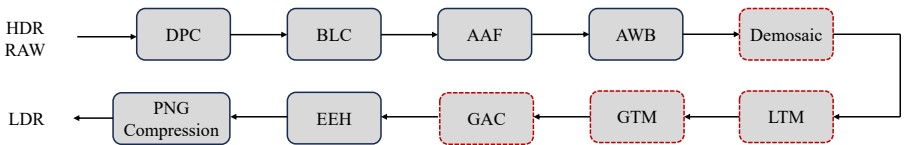

Figure 6: The key components of the HDR ISP pipeline, see text for details. We visualize the results for the modules marked with a red dotted line.

where $I_i$ is the synthetic exposure image, and $w$ represents the corresponding weights calculated based on image content.

**Global Tone Mapping**: The GTM (Global Tone-Mapping) block follows the LTM, enhancing the overall luminance. An S-shaped contrast-enhancing tone curve [13] is applied to the linear sRGB image. This curve is then concatenated with the sRGB color component transfer function, which transforms the image from linear sRGB to nonlinear sRGB. The Reinhard tone curves [13] can be expressed as follows:

$$\bar{I}_w = \frac{1}{N} \exp\left(\sum \log(\delta + I)\right) \tag{16}$$

$$I_{\text{gtm}} = \frac{I \cdot \left(1 + \frac{I}{\bar{I}_w^2}\right)}{1 + I} \tag{17}$$

where $\delta$ is a small value to avoid numerical overflow.

**Gamma Correction**: The GAC (gamma correction) is used to match the non-linear characteristics of a display device or human perception. We adopt the correction function Eq. (18) recommended in ITU-R BT. 709 standard [57], which is widely used in commodity cameras today.

$$I_{gamma} = \begin{cases} 12.92 \cdot I, & I \le 0.00304, \\ 1.055 \cdot I^{1/2.4} - 0.055, & I > 0.00304. \end{cases} \tag{18}$$

**Edge Enhancement**: The EEH (Edge Enhancement) module enhances image details and edges, improving image clarity and visual appeal. It is particularly useful for accentuating finer image structures. The module can be expressed as follows:

$$I_{\text{sharpen}} = p_s \cdot I + (1 - p_s) \cdot I_{\text{blurred}}, \tag{19}$$

In the ISP pipelines, these modules are combined in series and executed sequentially on the HDR RAW image to stably generate pleasing and consistent LDR sRGB images.

## B The Proof of the Approximation to Dynamic Range

1. Dynamic Range Ratio for a Single Gaussian Component. For a Gaussian component with mean $\mu$ and standard deviation $\sigma$, define the dynamic range:

$$d = \frac{\mu + \sigma}{\mu - \sigma}, \mu > \sigma. \tag{20}$$

This dynamic range measures the distance of the component relative to its mean.

2. Relating $d$ to $\mu$ and $\sigma$. Solve for $\mu$ in terms of $d$ and $\sigma$:

$$\mu = \sigma \cdot \frac{d+1}{d-1}. \tag{21}$$

3. Second Moment and Mean of a Single Component For a Gaussian component:

$$\mathbb{E}(x^2) = \sigma^2 + \mu^2, \quad \mathbb{E}(x) = \mu.$$

Thus:

$$\mathbb{E}(x^2) + \mathbb{E}(x) = \sigma^2 + \mu^2 + \mu.$$

Substitute $\mu = \sigma \cdot \frac{d+1}{d-1}$:

$$\mathbb{E}(x^2) + \mathbb{E}(x) = \sigma^2 + (\sigma \cdot \frac{d+1}{d-1})^2 + \sigma \cdot \frac{d+1}{d-1}. \tag{22}$$

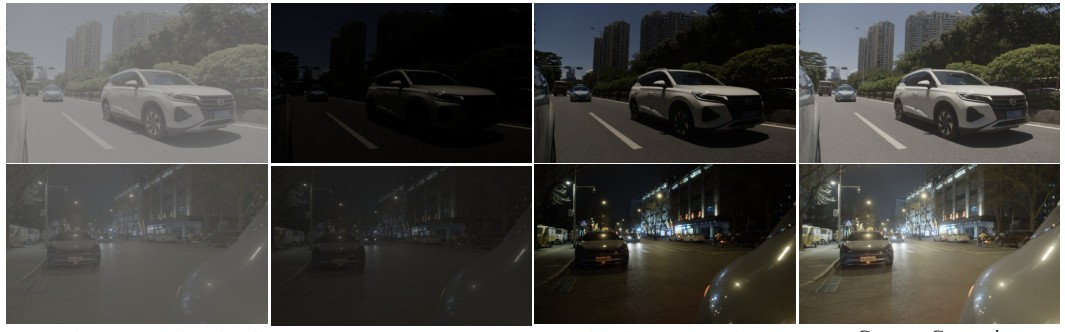

| Demosaic (Gamma for visualize) | Local Tone Mapping | Global Tone Mapping | Gamma Correction |

Figure 7: Visual comparison of key steps in the HDR ISP pipeline.

Simplify:

$$\mathbb{E}(x^2) + \mathbb{E}(x) = \sigma^2 \left[ 1 + \frac{(d+1)^2}{(d-1)^2} \right] + \sigma \cdot \frac{d+1}{d-1}. \tag{23}$$

4. Bounding $\mathbb{E}(x^2) + \mathbb{E}(x)$ using $R$ Expand the squared term

$$\frac{(d+1)^2}{(d-1)^2} = \frac{d^2 + 2d + 1}{d^2 - 2d + 1} = 1 + \frac{4d}{(d-1)^2}. \tag{24}$$

Substitute back

$$\mathbb{E}(x^2) + \mathbb{E}(x) = \sigma^2 \left[ 2 + \frac{4d}{(d-1)^2} \right] + \sigma \cdot \frac{d+1}{d-1} \tag{25}$$

5. Inequality Analysis Using the AM-GM (Arithmetic and Geometric Means) Inequality:

$$\frac{\mu + \sigma}{2} \geq \sqrt{\mu\sigma} \implies \mu + \sigma \geq 2\sqrt{\mu\sigma}. \tag{26}$$

For the dynamic range $d$:

$$d = \frac{\mu + \sigma}{\mu - \sigma} \geq \frac{2\sqrt{\mu\sigma}}{\mu - \sigma}. \tag{27}$$

This implies that $d$ grows as the overlap between $\mu$ and $\sigma$ increases, amplifying $\mathbb{E}(x^2) + \mathbb{E}(x)$.
6. For a GMM with $K$ components:

$$\mathbb{E}(x^2) + \mathbb{E}(x) = \sum_{i=1}^{K} \pi_i (\sigma_i^2 + \mu_i^2 + \mu_i). \tag{28}$$

Expressing each $\mu_i$ in terms of $d_i = \frac{\mu_i + \sigma_i}{\mu_i - \sigma_i}$:

$$\mathbb{E}(x^2) + \mathbb{E}(x) = \sum_{i=1}^{K} \pi_i \left[ \sigma_i^2 + \left( \sigma_i \cdot \frac{d_i + 1}{d_i - 1} \right)^2 + \sigma_i \cdot \frac{d_i + 1}{d_i - 1} \right]. \tag{29}$$

Ignoring first-order terms and constants and focusing on the dominant components, we obtain the final expression:

$$\mathbb{E}(x^2) + \mathbb{E}(x) \propto \sum_{i=1}^{K} \pi_i \left( \frac{d_i + 1}{d_i - 1} \right)^2, \quad d_i = \frac{\mu_i + \sigma_i}{\mu_i - \sigma_i} \tag{30}$$

## C  Detailed Proof of Scaling-Invariant Tone Mapping

We construct the tone mapper $\text{TM}_{\text{SI}}$ by a neural network composed of {*Conv-BN-ReLU*} with $L$ layers and remove all bias terms within this network. Then we start proving the TM is functionally

Table 7: Performance comparison with the radiance range of neural photometric on RoD Dataset. **Bold** denotes the default setting.

| Radiance Range | [1e3, 1e6] | [1e3, 1e7] | [1e3, 1e8] | **[1e4, 1e7]** | [1e4, 1e8] | [1e5, 1e7] | [1e5, 1e8] |
|---|---|---|---|---|---|---|---|
| mAP | 49.7 | 49.8 | 49.9 | **49.8** | 49.7 | 49.6 | 49.6 |
| mAR | 58.6 | 58.7 | 58.7 | **58.7** | 58.7 | 58.6 | 58.5 |

Table 8: Quantitative comparison of different pretraining losses.

| Method | Pretrained Loss | mAP | AP50 | AP75 | contrast |
|---|---|---|---|---|---|
| Faster R-CNN [4] | L1 | 41.3 | 67.1 | 48.2 | 0.08 |
| | NLPD [53] | **49.8** | **73.3** | **55.6** | **0.411** |

equivalent to a local tone mapping operator.

**1. Convolution**: For input $x$ and kernel $K_i$:

$$\text{conv}_i(\alpha x) = \alpha \cdot \text{conv}_i(x), \quad \text{where conv}_i(x) = K_i * x. \tag{31}$$

**2. Batch Normalization**: For input $x$ and kernel $K_i$:

$$\text{BN}_i(x) = \gamma_i \cdot \frac{x - \mu_i(x)}{\sigma_i(x)}, \tag{32}$$

where $\mu_i(\alpha x) = \alpha \cdot \mu_i(x)$ and $\sigma_i(\alpha x) = \alpha \cdot \sigma_i(x)$.

**3. ReLU**: For input $x$:

$$\text{RELU}(x) = \max(x, 0). \tag{33}$$

Then we start proving this bias-free network is scaling-invariant:

$$
\begin{aligned}
\text{TM}_{\text{SI}}(\alpha \cdot x) &= \text{RELU} \circ \text{BN}_L \circ K_L * \cdots \circ \text{RELU} \circ \text{BN}_1 \circ K_1 * (\alpha y) \\
&= \text{RELU} \circ \text{BN}_L \circ K_L * \cdots \circ \text{RELU} \circ \text{BN}_1 \circ (\alpha \cdot K \cdot y) \quad &\text{\#Convolution Linearity} \\
&= \text{RELU} \circ \text{BN}_L \circ K_L * \cdots \circ \text{RELU} \left( \gamma_1 \cdot \frac{\alpha \cdot K_1 * x - \alpha \cdot \mu_1}{\alpha \cdot \sigma_1} \right) \quad &\text{\#BN statistic scaling} \\
&= \text{RELU} \circ \text{BN}_L \circ K_L * \cdots \circ \text{RELU} \left( \gamma_1 \cdot \alpha \cdot \frac{K_1 * x - \mu_1}{\sigma_1} \right) \\
&= \text{RELU} \circ \text{BN}_L \circ K_L * \cdots \circ \alpha \cdot \text{RELU} \left( \gamma_1 \cdot \frac{K_1 * x - \mu_1}{\sigma_1} \right) \quad &\text{\#Homogeneity} \\
&= \alpha \cdot \text{RELU} \circ \text{BN}_L \circ K_L * \cdots \circ \text{RELU} \circ \text{BN}_1 \circ K_1 * x \\
&= \alpha \cdot \text{TM}_{\text{SI}}(x).
\end{aligned}
\tag{34}
$$

where $\circ$ denotes the cascading of network layers. In the scaling-invariant transformation, all operators apply a linear transformation on local neighborhoods. Here, $\text{BN}_i$ adaptively adjusts gains using local statistics ($\mu_i$, $\sigma_i$), and the cascade of $\text{BN}_i$ and ReLU activation mimics tone curves that compress highlights and shadows while preserving tones. Since all these linear transformations are applied within a local window, they effectively function as local tone mapping.

## D  More Ablation Studies

**Ablation of Radiance Range.** In the proposed neural photometric calibration, we set the radiance range as a hyperparameter. We conduct an ablation study on radiance and detection performance, with the results shown in Table 7. The findings demonstrate that our neural photometric calibration is not sensitive to the radiance range, highlighting the method's robustness in handling varying lighting conditions.

**Ablation of Pretraining Loss.** We conduct an ablation study on pretraining loss to demonstrate its effect on convergence performance. We compare the pretraining loss between NLPD [53] and L1 loss, with the results shown in Table 8. The L1 loss is supervised by HDR ISP results. The experiment shows that NLPD pretraining enables the tone mapper to enhance details, thereby improving detection performance.

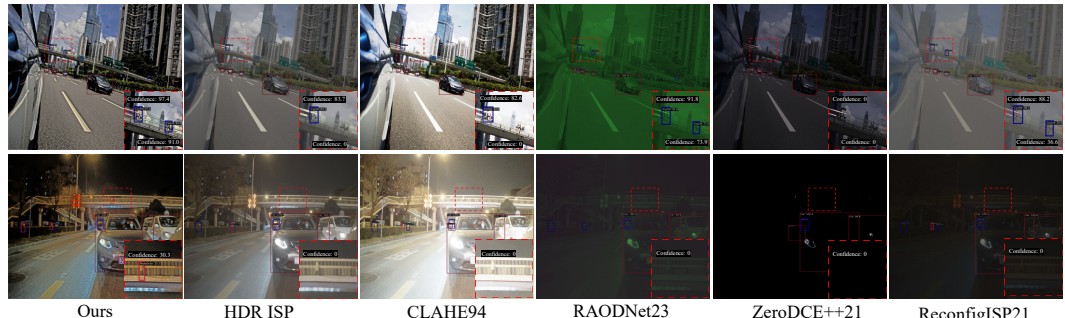

| Ours | HDR ISP | CLAHE94 | RAODNet23 | ZeroDCE++21 | ReconfigISP21 |

Figure 8: Visual comparison of different methods on HDR RAW inputs. The first row shows day scenes, while the second row presents night scenes. Our method outperforms the comparison methods. Please zoom in for confidence scores and class predictions.

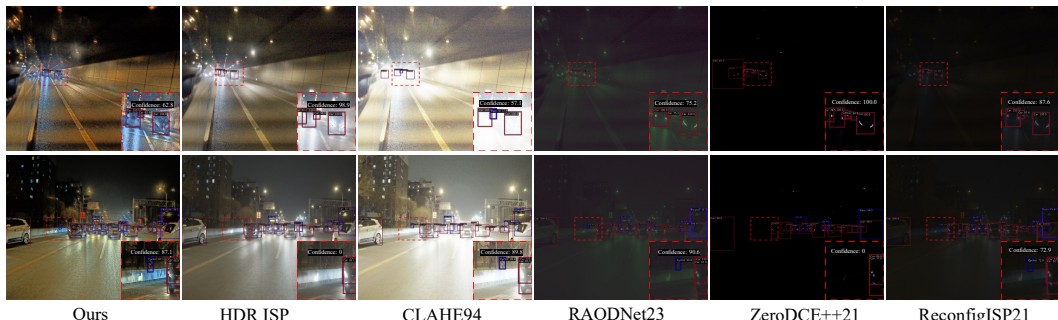

| Ours | HDR ISP | CLAHE94 | RAODNet23 | ZeroDCE++21 | ReconfigISP21 |

Figure 9: Visual comparison of different methods on HDR RAW inputs. The first row shows day scenes, while the second row presents night scenes. Our method outperforms the comparison methods. Please zoom in for confidence scores and class predictions.

## E    More Visual Comparison.

We show more visual results in Fig. 8 and Fig. 9. Specifically, we visualize the detection results of the comparison methods with confidence scores greater than 0.3, in different scenarios of the RoD dataset [15]. In the first row of Fig. 8, the tunnel environment is shown, where our method effectively reduces false detections. In Fig. 9, shows a typical HDR scene, where our method detects small objects that other methods fail to identify.

## F    Real World Evaluation.

**HDR Video Validation.** We collect HDR RAW video sequences from autonomous driving scenes to validate the proposed method, using YOLOv3 [5] as the base detector for this validation. The entire pipeline is evaluated on the NVIDIA Jetson AGX Orin (16-bit float) with 2K resolution ($2048 \times 1080$) input. The HDR RAW input is initially resized to a 4K resolution ($4096 \times 2160$) for processing. Additionally, we have created a **video demo (video_demo.mp4)** in the supplementary file to showcase the detection results on the video sequences. Please refer to the attachment. We will release the video sequences and corresponding annotations once the dataset is complete.

