# OpenReview forum: "Real-Time Scene-Adaptive Tone Mapping for High-Dynamic Range Object Detection"
_NeurIPS.cc/2025/Conference — NeurIPS 2025 poster_

### Official Review · Reviewer_zmDQ · 2025-06-19

**Clarity:** 3
**Significance:** 3
**Originality:** 3
**Rating:** 5
**Confidence:** 4

**Summary:**

The paper proposes a real-time scene-adaptive tone mapping method. Its core consists of a neural photometric calibration module for dynamically regularizing extreme dynamic ranges, and a scale-invariant local tone mapper that ensures linear response of mapping functions to input luminance scales, enhancing detail preservation and detection performance. Through end-to-end joint training optimization, this method improves the accuracy and generalization capability of HDR object detection.

**Questions:**

- If the dynamic range of HDR data is proportional to the square of the gradient magnitude, then why does training a detector based on HDR inputs lead to unstable gradient propagation, while training an HDR-to-LDR network does not exhibit such issues? Has there been any experimental analysis on the training stability with HDR inputs?

- Neural Photometric Calibration is performed in a low-resolution image space, which means that the predicted weight map is patch-wise. Approximately how large is each patch? Would this patch-wise weight map affect the scale-invariance of subsequent processing?

- The scale-invariant local tone mapping network proposed in the paper can effectively preserve the texture details of the image, while HDRNet is also a commonly used local tone mapping technique. The authors may consider comparing the differences between these two methods.

**Ethical Concerns:**

["NO or VERY MINOR ethics concerns only"]

**Final Justification:**

The author's rebuttal has addressed most of my concerns.

**Limitations:**

yes

**Paper Formatting Concerns:**

In Figure 2: "Conventioanl" → "Conventional"

**Quality:**

3

**Strengths And Weaknesses:**

- The paper  is clearly written and easy to comprehend.

- The paper proposes an innovative neural photometric calibration module along with a scale-invariant tone mapping network, which effectively enhances detection performance.

- In detection experiments under HDR scenarios, it also demonstrates outstanding performance, significantly surpassing existing state-of-the-art techniques.

---

> ### Author Rebuttal · Authors · 2025-07-30
>
> Dear Reviewer zmDQ,
>
> We'd like to thank you for your positive feedback and address the concerns raised in your comments.
>
>
>
> ## Concerns
>
> ## Q1. Why does training a detector based on HDR inputs lead to unstable gradient propagation, while training an HDR-to-LDR network does not exhibit such issues?
>
> 1.The 24-bit HDR (140dB) images exhibit an extreme dynamic range compared to 12-bit LDR images (70dB), with a ratio of approximately 10^7.
> Therefore, directly feeding HDR images into the network can result in extreme gradient fluctuations, potentially causing the loss function to become **NaN**.
>
> 2.Both HDR-to-LDR models and handcrafted tone mapping algorithms can compress the dynamic range while retaining as many details as possible, helping to stabilize the training process. The difference lies in the fact that the effectiveness of HDR-to-LDR models is influenced by network design and the training process. This is why we pretrain the tone mapper network initially. Traditional tone mapping algorithms, on the other hand, do not require pretraining of parameters, allowing them to converge stably.
>
>
>
> ## Q2.Has there been any experimental analysis on the training stability with HDR inputs?
>
> We use a reinahrd tone mapping [1] method  to compress the dynamic range and quantize it into different bit-depths before feeding it into the detector.
> Experiments show that 14-bit data yields the best performance, while 8-bit data yields the worst.
> A wider dynamic range preserves more useful details, but under extreme lighting conditions, it can render most details indistinguishable, making it challenging for neural networks to extract meaningful feature.
>
> | Input  | mAP   | AP50 | AP75 |
> | ------ | ----- | ---- | ---- |
> | 24-bit | —NAN— |      |      |
> | 14-bit | 48.1  | 69.4 | 51.4 |
> | 12-bit | 47.8  | 69.0 | 47.8 |
> | 8-bit  | 43.8  | 65.6 | 45.8 |
>
>
>
>
>
> ## Q3. What is the size the predicted weight map in low-resolution ? Would this patch-wise weight map affect the scale-invariance of subsequent processing?
>
> Our method predicts the scale map $K$ in a low-resolution space (256x256) and then upsamples it to the original resolution using bilinear interpolation.
> This scale map  does not affect the scale-invariant property of the tone mapper, which is determined by the network design.
> Bias-free CNNs (with BN and ReLU) generally exhibit scale invariance.
>
>
>
>
>
> ## Q4. The authors may consider comparing with HDRNet.
>
> We compared HDRNet[2] and the results are as follows.
>
> | Pipeline                       | mAP   | AP50 | AP75 | AR75 |
> | ------------------------------ | ----- | ---- | ---- | ---- |
> | HDRNet (from scratch)          | —NAN— |      |      |      |
> | HDRNet (pretrained from MIT5K) | 41.9  | 64.9 | 46.1 | 49.2 |
> | Our Method                     | 49.8  | 73.3 | 55.6 | 58.7 |
>
>
> HDRNet (from scratch) refers to a cascade with a detector trained from scratch, while HDRNet (pretrained) is pretrained on the RoD dataset (HDR-LDR pairs).
>
>
>
>
>
>
> [1] Reinhard, Erik, et al. "Photographic tone reproduction for digital images." Seminal Graphics Papers: Pushing the Boundaries, Volume 2. 2023. 661-670.
>
> [2] Gharbi, Michaël, et al. "Deep bilateral learning for real-time image enhancement." ACM Transactions on Graphics (TOG) 36.4 (2017): 1-12.

---

> > ### Comment · Reviewer_zmDQ · 2025-08-05
> >
> > Regarding question 1, I still have some concerns about the distinction between these two training approaches, as both employ backpropagation and objective function optimization. If specific network architectures and training configurations can stabilize the training process for HDR-to-LDR models, then they should theoretically also be capable of stabilizing the detector's training.
> >
> > Regarding question 2, while the authors demonstrate that HDR inputs can enhance detector performance, what I'm more interested in understanding is the analysis of training stability.

---

> > > ### Author Response · Authors · 2025-08-06
> > >
> > > Thanks for the discussion, I think we need to clarify some concepts at first.
> > >
> > >
> > > ## Q1.
> > >
> > > Since **24-bit HDR** images have an extreme dynamic range, valid image details in the dark regions are often hidden. Therefore, tone mapping is required as a preprocessing step before feeding the image into the detector.
> > > In Q1, we discuss two approaches: **the DNN-based tone mapping model** (HDR-to-LDR model) and **the handcrafted tone mapping algorithm**.
> > > The handcrafted tone mapping algorithm is designed based on human visual perception.  However, this approach is suboptimal for object detection tasks, and since these algorithms are not learnable, they cannot be jointly trained with the object detector.
> > >
> > >
> > >
> > >
> > >
> > > The DNN-based tone mapping model (HDR-to-LDR net) can be integrated with object detection training, enabling it to achieve optimal performance.
> > >
> > > It is important to note that the joint training is different from training the tone mapper separately.
> > > Tone mapping is typically trained on HDR-LDR pairs using L1 Loss, whereas joint training is driven by detection loss, with no intermediate supervision for the tone mapper.
> > >
> > > As a result, tone mappers must be suitably designed to stabilize gradient propagation. Some tone mappers with more complex designs lead to performance degradation (such as AnscombeNet21 in Tab. 1 and HDRNet).
> > > The proposed tone mapper only uses convolution, batch normalization (BN), and ReLU. This simple design promotes effective gradient propagation and ensures scale invariance.
> > >
> > >
> > >
> > > ## Q2.
> > >
> > > We provide an analysis of dynamic range and gradients in the paper, demonstrating that a **higher dynamic range** can lead to **increased gradient fluctuations** (Eq. 2 in the paper).
> > > Therefore, a **tone mapper** is necessary to compress the dynamic range while retaining as much image detail as possible.
> > > Therefore, we provide the **LDR Image metrics** generated by different tone mappers, the results in Tab. 3.
> > > Images with greater contrast and entropy contain more details, which stabilizes the training stability and improves the benefits of the detector.
> > >
> > >
> > >
> > > If you have any questions, please feel free to continue the discussion with me.

---

> > > > ### Author Response · Authors · 2025-08-06
> > > > **Analysis of Training Stability**
> > > >
> > > > Thanks for your suggestions, we attempt to extend our proof and model the training stability using the dynamic range and gradient variance. We will include this derivation in the supplementary material.
> > > >
> > > >
> > > >
> > > > Firstly, we start from Eq. (2) in the paper and define the variance of the gradient to denote stability:
> > > > $$
> > > > \frac{\partial L}{\partial \omega} \propto \sum_{i=1}^K (\frac{d_i+1}{d_i-1})^2
> > > > $$
> > > >
> > > > $$
> > > > \frac{\partial L}{\partial w}=-2(y-\hat{y})x.
> > > > $$
> > > >
> > > > Then we derivate the variance :
> > > > $$
> > > > \text{Var}(\frac{\partial L}{\partial \omega})=4\text{Var}\left((y-\hat{y})x\right)=4[\mathbb{E}((y-\hat{y})^2x^2)-(\mathbb{E}((y-\hat{y})x))^2]
> > > > $$
> > > > Assuming that the residual $\epsilon=y-\hat{y}$ is independent of $x$, and that the model is such that $\mathbb{E}[\epsilon]=0$ and  $\text{Va}(\epsilon)=c^2$(constant), then
> > > > $$
> > > > \text{Var}\left(\frac{\partial L}{\partial w}\right)=4c^2\mathbb{E}[x^2]
> > > > $$
> > > >
> > > >
> > > > We assume that, under optimal conditions, each component of the GMM model has a similar variance to simplify the derivation,  $\sigma=\sigma_i$ for all $i$. Then,
> > > > $$
> > > > \mathbb{E}[x^2]=2\sigma^2\sum_{i=1}^K\pi_i\frac{d_i^2+1}{(d_i-1)^2}
> > > > $$
> > > > Then the gradient variance becomes:
> > > > $$
> > > > \mathrm{Var}\left(\frac{\partial L}{\partial w}\right)=4c^2\cdot2\sigma^2\sum_{i=1}^K\pi_i\frac{d_i^2+1}{(d_i-1)^2}=8c^2\sigma^2\sum_{i=1}^K\pi_i\frac{d_i^2+1}{(d_i-1)^2}
> > > > $$
> > > > The $c^2\sigma^2$ is a constant for given image, so we can ignore this term and obtain a proportional relationship to:
> > > > $$
> > > > \mathrm{Var}\left(\frac{\partial L}{\partial w}\right)\propto\sum_{i=1}^K\pi_i\left(\frac{d_i^2+1}{(d_i-1)^2}\right)
> > > > $$
> > > >
> > > >
> > > > From the final equation, we observe that when a single component covers the entire histogram, i.e., as $d_i\rightarrow 1^+$,  $\text{var} \rightarrow \infty$.
> > > >
> > > > This means that a **large dynamic range** in the input image can lead to infinite variance, which causes the **exploding gradient**.

---

> > > > > ### Author Response · Authors · 2025-08-07
> > > > >
> > > > > Dear Reviewer zmDQ,
> > > > >
> > > > >
> > > > > Thank you very much for your encouraging follow-up. We are grateful that you found many of our revisions satisfactory and that you have maintained a positive overall assessment.
> > > > > We take your remaining concerns seriously regarding training stability, advanced tone mapper design and lightweight computational strategies, and will explore them in future works.
> > > > > Your constructive comments continue to guide our work, and we appreciate the opportunity to improve the manuscript.
> > > > >
> > > > >
> > > > >
> > > > > Best,
> > > > >
> > > > >
> > > > > Authors of paper #14490

---

> ### Author Response · Authors · 2025-08-09
> **Rebuttal Summary**
>
> Dear Reviewer zmDQ
>
> The discussion period is ending soon. We hope this discussion has addressed your concern regarding **training stability**.
>
> During the rebuttal phase, we used **gradient variance**, to represent **training stability** and modeled the relationship between gradient variance and dynamic range. We modeled the dynamic range using a histogram that spans from 0 to 1, such that the dynamic range $d_i=\frac{\mu_i+\sigma_i}{\mu_i-\sigma_i}>1$. Quantitative studies show that as $d_i$ increases, gradient variance also increases, leading to training instability.
>
> I hope these responses address your concerns and provide some insights.
>
> Best,
>
> Authors

---

> > ### Comment · Reviewer_zmDQ · 2025-08-09
> >
> > Thank you to the author for responding to my question. I find this topic very interesting.
> >
> > Based on my understanding, the author highlighted the following points in the rebuttal: To address the issue where directly inputting HDR images into neural networks can lead to unstable gradient propagation and increased training difficulty, they designed a tone mapper with both unbiased characteristics and scale invariance. This carefully designed module not only effectively stabilizes the training process but also significantly enhances image detail representation through additional tone mapping.
> >
> > I am curious, in the design of detectors, could adopting this concept of unbiasedness and scale invariance similarly improve the stability of detector training? Specifically, is a scale-invariant network architecture the key factor for achieving stable training with high dynamic range data?

---

> > > ### Author Response · Authors · 2025-08-09
> > >
> > > Dear Reviewer zmDQ
> > >
> > > **Regarding the tone mapper:**
> > >
> > > To be precise, we designed a tone mapper with both **local transformation** **characteristics** and **scale invariance**.
> > >
> > > Scale invariance is a crucial requirement for HDR-to-LDR mapping, and a bias-free network can achieve this property. Local transformation can efficiently compress the dynamic range compared to global transformations (such as gamma compression).To achieve these two properties, we use a bias-free CNN as the tone mapper, which is a convenient design without too much Inductive bias.
> > >
> > > The local transformation property significantly enhances image detail representation, while scale invariance stabilizes the training process and improves generalization (Including generalization within a dataset and generalization across datasets.).
> > >
> > >
> > >
> > > **Regarding the design of detector**:
> > >
> > > Scale-invariant networks were initially applied to denoising tasks [1] and later extended to image restoration tasks [2]. What is certain is that scale invariance has been shown to enhance the performance of image-to-image tasks.
> > >
> > >
> > >
> > > Scale-invariant network architectures enable compressing the dynamic range, thereby stabilizing the training process for HDR input. For detector design, we can modify the backbone network to incorporate scale invariance, naturally enabling it to handle HDR inputs.
> > >
> > > This idea is highly valuable and could even be extended to detection on noisy and other low-quality images.
> > >
> > >
> > >
> > >
> > >
> > > [1] Mohan, Sreyas, et al. "Robust And Interpretable Blind Image Denoising Via Bias-Free Convolutional Neural Networks." *8th International Conference on Learning Representations, ICLR 2020*. 2020.
> > >
> > > [2] Zamir, Syed Waqas, et al. "Restormer: Efficient transformer for high-resolution image restoration." *Proceedings of the IEEE/CVF conference on computer vision and pattern recognition*. 2022.

---

### Official Review · Reviewer_2qV1 · 2025-07-01

**Clarity:** 3
**Significance:** 3
**Originality:** 3
**Rating:** 5
**Confidence:** 4

**Summary:**

This paper introduces a novel tone mapping framework that bridges the gap between HDR RAW inputs and LDR sRGB requirements for deeplearning alghrithoms, enabling end-to-end optimization with the object detection task.
The proposed approach replaces traditional ISP pipelines with neural photometric calibration and a scaling-invariant local tone mapping module to regulate dynamic ranges while preserving details.
It outperforms existing tone mapping and AI-ISP methods.

**Questions:**

See Weaknesses.

**Ethical Concerns:**

["NO or VERY MINOR ethics concerns only"]

**Final Justification:**

The author's response addressed my concerns, and I will maintain my original rating.

**Limitations:**

Yes.

**Quality:**

3

**Strengths And Weaknesses:**

Strengths:
1. The proposed framework jointly optimizes tone mapping and downstream detection tasks with improved performance over traditional ISP pipelines and existing improvement strategies.
2. Fine-tuning from LDR to HDR models can be supported at a much lower computational cost, increasing the practicality for real-world applications.
3. The efficiency of the proposed framework ensures high speed performance on embedded hardware suitable for autonomous driving applications.

Weaknesses:
1. Why is the learnable transformation function in Equation 4 designed in this particular form?
2. There is some ambiguity in the model description regarding the use of "Tone Mapper." From the contributions, Figure 2, and Section 3.3, it appears that the Tone Mapper is only one component of the proposed tone mapping framework, working alongside neural photometric calibration. However, in Sections 3.4 and 4.3 discussing model optimization, "Tone Mapper" seems to refer collectively to both components.
3. In the experiments of Table 5, how exactly was the scaling-invariant property removed?
4. In Table 1, different algorithms should be clearly labeled with their respective grouping characteristics to better highlight the focus of each algorithm category.

---

> ### Author Rebuttal · Authors · 2025-07-30
>
> Dear Reviewer 2qV1,
>
> We'd like to thank you for your positive feedback and address the concerns raised in your comments
>
>
>
> ## Concerns
>
> ## Q1. Why is the learnable transformation function in Equation 4 designed in this particular form?
>
> Since the input HDR RAW images are recorded as digital numbers, we need to make educated guesses about the minimum and maximum radiance values in the original scene. We introduce Eq. 4, where the bias term is constant to avoid excessively small values, and the scale map consists of learnable parameters that adjust the dynamic range. This approach maps HDR inputs into a unified luminance space through neural photometric calibration, addressing the large diversity in lighting conditions (e.g., day/night, mixed scenes).
>
>
>
> ## Q2. There is some ambiguity in the model description regarding the use of "Tone Mapper."
>
> Thank you for your suggestion. We will further clarify the description and distinguish the details between the tone mapper and neural photometric calibration in the paper. The tone mapping framework includes the scale-invariant tone mapper and neural photometric calibration, which work together for object detection. In Section 3.4, we pretrain only the scale-invariant tone mapper to warm up the parameters, enabling the tone mapper to gain the ability to compress the dynamic range. In Section 4.3, under the performance transfer strategy, we fine-tune the parameters of the tone mapping framework (tone mapper + neural photometric calibration) and the detection head to achieve rapid adaptation from the LDR RGB domain to the HDR RAW domain.
>
>
>
> ## Q3. In the experiments of Table 5, how exactly was the scaling-invariant property removed?
>
> Scale invariance means that the output changes in the same scale as the input. For HDR-to-LDR tone mapping, we argue that if the tone mapping network operates by scaling the HDR image onto a linear space of LDR images, then that space should encompass all rescalings of those images, especially including the origin. To achieve this, we design a bias-free CNN where, for any input $x$, the components (convolution, ReLU, and batch normalization) satisfy $f(ax)=af(x)$ for $x>0$. This ensures that any negative entries in the input are set to zero. Multiplying by a nonnegative constant does not change the sign of the entries in a vector.
> Thus, the bias-free network remains scale-invariant and can rescale the image without introducing any bias value. In contrast, a bias network would introduce additional background noise in the resulting images.
>
>
>
> ## Q4. In Table 1, different algorithms should be clearly labeled with their respective grouping characteristics to better highlight the focus of each algorithm category.
>
>
>
> Thank you for your suggestion. We will add the category group and modify Table 1 to the following style.
>
> | Method Group                | Methods       |
> | --------------------------- | ------------- |
> | Direct                      | HDR RAW       |
> | ISP Pipeline                | HDR ISP       |
> | Tone Mapping                | Mantiuk08     |
> |                             | CLAHE94       |
> | Differentiable ISP Pipeline | AnscombeNet21 |
> |                             | ReconfigISP21 |
> | Low-Light Enhance           | SCI22         |
> |                             | Zero-DCE++21  |
> | Differentiable Tone Mapping | IANet22       |
> |                             | RAODNet23     |
> |                             | RawOrCooked23 |
> |                             | Ours          |

---

### Official Review · Reviewer_4R6p · 2025-07-01

**Clarity:** 3
**Significance:** 3
**Originality:** 2
**Rating:** 4
**Confidence:** 4

**Summary:**

This work presents a method for HDR tonemapping as an input to detection networks. The authors present their method as an ISP-like approach that processes HDR captures fed into a pre-trained detection network. The method is evaluated on existing HDR datasets and using a prototype system that runs at real-time rates. The evaluation is focused on assessing various ISP and pre-processing methods that confirm the approach achieves consistent improvement over existing methods.

**Questions:**

Please comment on the finetuning of the detection network and missing RAW evaluations as discussed above.

**Ethical Concerns:**

["NO or VERY MINOR ethics concerns only"]

**Final Justification:**

I have increased my rating as the rebuttal addressed my concerns around major gaps in the evaluation. That said, concerns around novelty and technical contribution still remain, as also acknowledged by the authors (core contribution being a scale-invariant tonemapper).

**Limitations:**

Yes

**Quality:**

3

**Strengths And Weaknesses:**

Strengths:

- The pre-processing module (the ISP) is evaluated and compared to existing baseline methods in detail.
- The method is described in detail, and parameter settings are evaluated adequately.
- The approach is implemented on a realt-time platform.

Weaknesses:

- It is unclear if the evaluation is fair. My main concerns is here that the authors have not commented on whether and how they fine-tune the downstream network. As they already use detection losses, it remains unclear if the network has been finetuned here.
- As such, baseline evaluations without any ISP (finetuned on RAW data) are missing.
- The tonemapper is a convolutional network with scale invariant BN, which offers little technical novelty.

As such, overall, without extensive evaluations on fine-tuning I remain on the fence in this stage of the review process.

---

> ### Author Rebuttal · Authors · 2025-07-30
>
> Dear Reviewer 4R6p,
>
> Thank you very much for your insightful and valuable feedback. We believe there might be some misunderstandings, and we sincerely hope to clarify them and adequately address your concerns.
>
> ## Misunderstandings
>
> ## Q1.  Evaluations without ISP (HDR RAW Data).
>
> Actually, we have reported the performances of using HDR RAW data directly and HDR ISP in **Table 1** and present the data point as follows.
>
> | Pipeline           | mAP   | AP50 | AP75 | AR75 |
> | ------------------ | ----- | ---- | ---- | ---- |
> | HDR RAW （Direct） | —NAN— |      |      |      |
> | HDR ISP            | 42.9  | 65.9 | 46.4 | 49.6 |
> | Our Method         | 49.8  | 73.3 | 55.6 | 58.7 |
>
> For HDR RAW data, we cannot get normal results (0.049, 0.092, 0.032, 0.059) for training collapse.  It is important to note that 24-bit HDR RAW data has an extremely large dynamic range.  Directly using HDR RAW data can lead to gradient explosion, causing the loss function to become **NaN**, which makes the training process **non-convergent**.
>
> Regarding HDR ISP, the results of other methods are significantly lower than those of our approach.
>
> Furthermore, Reviewer 2qV1 acknowledged that our framework demonstrates improved performance compared to traditional ISP pipelines. Reviewer zmDQ regarded our method as having outstanding performance, notably outperforming existing state-of-the-art techniques.
>
>
>
> ## Q2. Missing the details on whether and how they fine-tune the downstream network.
>
> In this paper, we cascade the proposed tone mapping framework with the object detection network. The detection network is initialized with COCO pretrained weights and jointly trained with the tone mapping framework using detection loss in an end-to-end manner. Apart from this, we did not perform any fine-tuning on the detection network.
>
> ## Concerns
>
> ## Q3. It is unclear if the evaluation is fair.
>
> Following the RoD dataset [1], we choose the RAODNet23 [1], IANet22 [2], which is most relevant works to ours
>
> For handcrafted tone mapping methods, we select Mantiuk08 [4] and CLAHE [5], which are included in popular development packages and widely used in applications.
>
> We also compare other types of comparative methods, including the Differentiable ISP Method, Low-Light Enhancement Method, and DNN-based ISP Network.
>
> Among these, RAODNet23 [1] serves as the strong baseline method for the RoD dataset, and our method shows significant improvement in comparison (+3.7mAP).
>
>
> ## Q4. The tone mapper offers little technical novelty.
>
> The innovativeness of our design is reflected in the following three aspects:
>
> 1. The structure we proposed possesses **scale-invariant** properties, which are supported by **mathematical theories**. Tone Mapping is employed to compress the dynamic range and preserve details when converting from HDR to LDR. Theoretically, tone mapping necessitates the capacity to map HDR to LDR on a per - pixel basis under arbitrary luminance conditions. This implies that tone mapping must consider all scaling factors, with particular inclusion of the origin (mapping any value to zero). This realization has inspired us to design a scale-invariant tone mapper by eliminating all bias terms. We have proven that bias-free CNNs are scale-invariant, as shown in Eq. 8, and are also equivalent to local tone mapping, as demonstrated in Eq. 24 (in the Appendix).
>
> 2. Compared with other methods, the design of the tone mapper is **efficient and concise**, eschewing complex predefined models. In contrast to related works such as [1], [2], and [3], these approaches pre - define tone curves and estimate curve parameters to dynamically compress the dynamic range for each input. Moreover, the most recent tone-mapping methods, like [4] and [5], utilize Look-Up Tables (LUTs) and pyramid decomposition. This results in a more complex pipeline that is not suitable for deployment on edge devices.
>
> 3. Our method holds significant **practical** value. It exhibits high-speed performance and incurs a lower computational cost. All reviewers, including you, have acknowledged that our method is highly suitable for real-world applications.
>
>
>
>
>
> [1] Xu, Ruikang, et al. "Toward raw object detection: A new benchmark and a new model." Proceedings of the IEEE/CVF conference on computer vision and pattern recognition. 2023.
>
> [2] Liu, Wenyu, et al. "Image-adaptive YOLO for object detection in adverse weather conditions." Proceedings of the AAAI conference on artificial intelligence. Vol. 36. No. 2. 2022.
>
> [3] Li, Chongyi, Chunle Guo, and Chen Change Loy. "Learning to enhance low-light image via zero-reference deep curve estimation." IEEE transactions on pattern analysis and machine intelligence 44.8 (2021): 4225-4238.
>
> [4] Jiang, Ting, et al. "Meflut: Unsupervised 1d lookup tables for multi-exposure image fusion." Proceedings of the IEEE/CVF International Conference on Computer Vision. 2023.
>
> [5]  Zhang, Feng, et al. "Lookup table meets local laplacian filter: pyramid reconstruction network for tone mapping." Advances in Neural Information Processing Systems 36 (2023): 57558-57569.

---

> > ### Comment · Reviewer_4R6p · 2025-08-05
> > **Q2 Finetuning**
> >
> > Thank you for the rebuttal. I am not following the clarification in Q2. Can you please clarify if the baseline (without your tonemapper) is finetuned or not?

---

> > > ### Author Response · Authors · 2025-08-06
> > >
> > > Thanks for the discussion. I think it's important to clarify some concepts.
> > >
> > > The **full parameter training strategy** trains **all parameters** for 13 epochs using COCO pretrained weights, aiming for higher performance.
> > > All **comparison experiments** (Tab 1 in paper) use the full parameter training strategy. These comparison methods take HDR RAW images as input, output detection results, and apply the full parameter training strategy to train both the tone mapper and the detector.
> > >
> > > The **performance transfer strategy** only **finetunes** the partial parameters (tone mapper + detector head,  accounting for <4% of the total parameters) for 5 epochs, utilizing existing LDR RGB pretrained weights to achieve performance transfer.
> > >
> > > In Tab 2, we also compare our method with the performance transfer methods [1] and [2],  and our approach outperforms the other related works.  And we copy the results as follows:
> > >
> > > | Pretrained Weights | Methods       | mAP      | mAR      |
> > > | ------------------ | ------------- | -------- | -------- |
> > > | COCO               | RAWAdapter24  | 12.6     | 25.5     |
> > > |                    | AdaptiveISP24 | 22.5     | 33.5     |
> > > |                    | Ours          | **30.6** | **47.5** |
> > > | Object365          | RAWAdapter24  | 15.7     | 28.9     |
> > > |                    | AdaptiveISP24 | 24.7     | 36.1     |
> > > |                    | Ours          | **38.8** | **50.7** |
> > >
> > >
> > >
> > >
> > >
> > > The **baseline (without tone mapper)** uses **HDR RAW data directly** as input and applies the **full parameter training strategy** to optimize the detection network parameters.
> > > Since 24-bit HDR RAW has an extreme dynamic range, image details in the dark regions are hidden, making it challenging for the neural network to extract effective features.
> > >
> > >
> > >
> > > If you have any questions, please feel free to continue the discussion with me.
> > >
> > >
> > >
> > > [1] Cui, Ziteng, and Tatsuya Harada. "RAW-adapter: Adapting pre-trained visual model to camera RAW images." European Conference on Computer Vision. Cham: Springer Nature Switzerland, 2024.
> > >
> > >
> > > [2] Wang, Yujin, et al. "Adaptiveisp: Learning an adaptive image signal processor for object detection." Advances in Neural Information Processing Systems 37 (2024): 112598-112623.

---

> > > > ### Comment · Reviewer_4R6p · 2025-08-06
> > > > **Unclear Baseline Evaluations**
> > > >
> > > > Dear Authors,
> > > >
> > > > this is not addressing my question. I am wondering if the models run on top of the outputs of HDR-ISP (Faster RCNN and Yolo) are fully finetuned on the data that HDR-ISP produces (reprocessed training dataset) or not. This is not clear in the manuscript. Also, is HDR-ISP reimplemented? If so, how?

---

> > > > > ### Author Response · Authors · 2025-08-06
> > > > > **For Baseline Evaluations**
> > > > >
> > > > > Thank you for response, we find the term "HDR-RAW" in Tab 1 leads some ambiguity, it is both data type and processing method here. And we will modify the table for clear understanding.
> > > > >
> > > > > **Baseline Evaluation:** Yes.
> > > > >
> > > > > In Tab. 1, the term "HDR-ISP" refers to processing the HDR-RAW image as RGB images, after which the object detection networks (Faster RCNN and Yolo) are fine-tuned on this reprocessed training dataset.
> > > > >
> > > > > Other comparison methods also process the input HDR-RAW image and feed the processed images into the object detection networks for fine-tuning.
> > > > >
> > > > > **HDR-ISP Implementation:** The HDR-ISP method is based on HDRPlus [1] [2], a professional ISP pipeline for HDR imaging. We reimplemented it to process the used dataset. We report the implementation details in paper (line248-250) and supplementary (sec 1).
> > > > >
> > > > >
> > > > >
> > > > > We will use **"RAW baseline"** to replace **"HDR-RAW"** in Tab. 1, which denotes taking HDR-RAW images directly into the object detection network without any processing.
> > > > >
> > > > >
> > > > >
> > > > > [1]  Hasinoff, Samuel W., et al. "Burst photography for high dynamic range and low-light imaging on mobile cameras." *ACM Transactions on Graphics (ToG)* 35.6 (2016): 1-12.
> > > > >
> > > > > [2] Monod, Antoine, Julie Delon, and Thomas Veit. "An analysis and implementation of the hdr+ burst denoising method." *Image Processing On Line* 11 (2021): 142-169.

---

> > > > > > ### Author Response · Authors · 2025-08-07
> > > > > > **Looking forward to your reply**
> > > > > >
> > > > > > Dear Reviewer 4R6p,
> > > > > >
> > > > > >
> > > > > > Thank you very much for your thoughtful and thorough review. We truly appreciate the time and effort you put into providing such valuable feedback.
> > > > > > We have added additional experimental results, provided detailed explanations, and offered comprehensive responses to each question in an effort to address your concerns.
> > > > > >
> > > > > >
> > > > > > We sincerely hope that these revisions have sufficiently addressed your concerns. If there are any remaining questions or areas requiring further clarification, please do not hesitate to let us know. We would be more than happy to provide additional details or experiments to ensure your satisfaction.
> > > > > > Once again, thank you for your time, consideration, and invaluable input.
> > > > > >
> > > > > >
> > > > > >
> > > > > > Best,
> > > > > >
> > > > > >
> > > > > > Authors of paper #14490

---

> ### Author Response · Authors · 2025-08-09
> **Rebuttal Summary**
>
> Reviewer 4R6p
>
> The discussion period is ending soon. We hope this discussion has addressed your concerns regarding **evaluation fairness** and **novelty**.
>
>
>
> Regarding **evaluation fairness**, we provide the implementation details in Section 4 of this paper. All comparison methods are treated as tone mappers, cascaded with the detector and trained end-to-end using a detection loss. The tone mapper preprocesses the input HDR-RAW image before feeding it to the detector network.
>
> The HDR-ISP method generates processed images offline before performing object detection. While the other methods run the detector jointly, processing images and applying detection online.
> We follow the standard strategy for detector training: "1x" schedule, meaning initialization with coco pretrained weights, 13 epochs of training, and multi-step learning decay. These settings are adopted from previous work.
>
> Regarding **novelty**,  we analyze the requirements of HDR tone mapping and design a tone mapper based on the principle of scale invariance. Although the proposed tone mapper is straightforward, it is highly efficient, balancing performance and speed without unnecessary complexity. It can process 4K resolution input in real-time, a capability that comparison methods cannot achieve.
>
>
>
> I hope these responses address your concerns and provide some insights.
>
> Best,
>
> Authors

---

### Official Review · Reviewer_Vk4C · 2025-07-01

**Clarity:** 2
**Significance:** 2
**Originality:** 2
**Rating:** 3
**Confidence:** 2

**Summary:**

The paper proposes a tone mapping method that can be jointly trained end-to-end with neural object detection models. The method first applies a neural photometric calibration module to handle varying lighting conditions, followed by a scale-invariant tone mapping module. The entire pipeline is optimized jointly with the object detector to improve detection performance under diverse illumination.

**Questions:**

- Could the authors clarify why joint training of the tone mapper and object detector is preferable compared to using a modular tone mapping approach followed by detection?
- Are there scenarios where the added complexity of joint training is justified, and if so, could the authors provide practical examples or empirical evidence illustrating these benefits?

I am not an expert in this specific area, so I am open to reconsidering my rating if there are domain-specific factors I may have overlooked.

**Ethical Concerns:**

["NO or VERY MINOR ethics concerns only"]

**Final Justification:**

I appreciate the additional experiments provided by the authors; however, they confirm my concerns about the lack of modularity. In my view, the need to jointly retrain the tonemapper for every new task introduces an additional level of complexity and is a clear disadvantage. Therefore, I will maintain my original rating.

**Limitations:**

-

**Paper Formatting Concerns:**

No formatting concerns

**Quality:**

2

**Strengths And Weaknesses:**

Strengths:
- The proposed method improves object detection performance when jointly trained with several commonly used detectors.
- The approach demonstrates low latency, making it practical for real-time deployment.

Weaknesses:
- Since the proposed tone mapping is specifically trained end-to-end with the detector, it is somewhat expected that detection performance would improve under this setup.
- It is unclear why this design choice is preferable, as joint end-to-end training sacrifices modularity and increases training pipeline complexity. This seems like a significant disadvantage that may outweigh the potential benefits of joint optimization, and the paper currently does not provide a clear justification for this trade-off.

---

> ### Author Rebuttal · Authors · 2025-07-30
>
> Dear Reviewer Vk4C,
>
> Thank you for your feedback. We’d like to provide some contextual information to help you better understand our method and address your concerns.
>
> ## Learning-based HDR Perception
> (1) HDR data contains richer tones and more details, which have the potential to boost various computer vision tasks. However, neural networks are usually designed for LDR input, which requires compressing the dynamic range before directly utilizing HDR data.
>
> (2) There are many ways to compress the dynamic range, with tone mapping being the most common. Handcrafted tone mapping methods are based on human visual perception, while deep learning-based tone mapping is guided by the loss function.
>
> ## Concerns
>
> ## Q1. Why is joint training of the tone mapper and object detector preferable compared to using a modular tone mapping approach followed by detection?
>
> The HDR images present an extreme dynamic range compared to LDR images. For example, 24-bit HDR contains about 1e7 times the dynamic range of 12-bit LDR, directly feeding these HDR images into neural network can lead to gradient explosion.
> Therefore, tone mapping is necessary to compress the dynamic range initially.
> Most modular tone mapping methods are crafted to adhere to human visual perception, but this alignment may not always be conducive to meeting the demands of downstream high-level tasks.
> Since a DNN-based tone mapper is cascaded with an object detector, which optimizes the model parameters driven by detection loss, it can generate the most suitable image for enhanced detection performance.
> Besides, some of these DNN-based tone mappers are plug-and-play (such as our method) and can be cascaded with any object detector without any internal changes, which can also be seen as a modular approach.
>
>
>
> ## Q2. Are there scenarios where the added complexity of joint training is justified, and if so, could the authors provide practical examples or empirical evidence illustrating these benefits?
> In challenging imaging conditions (e.g., high dynamic range scenarios, rainy/haze weather), target detection requires a specialized image preprocessor.
> Some handcrafted image preprocessing methods provide limited generality, as they cannot generalize well across all examples.
> To address this, cascading a DNN-based image preprocessor (eg, tone mapper,  dehaze network) with an object detector can improve performance, although it introduces additional complexity.

---

> > ### Comment · Reviewer_Vk4C · 2025-08-04
> >
> > Thank you for your responses. Perhaps my question regarding modularity was not totally clear, so let me clarify.
> >
> > What I am asking is essentially the following: suppose you have trained your tone mapper jointly with a specific object detector (say Faster R-CNN). Later, you want to test another detector (say Deformable DETR).
> >
> > Would you need to **jointly** retrain with Deformable DETR ? How is the detection performance affected if the tone mapper is not retrained (are there any related quantitative results in the paper that I might have missed) ?

---

> > > ### Author Response · Authors · 2025-08-05
> > >
> > > ## Q1: Testing with a pretrained tone mapper and another detector
> > > In general, retraining is necessary because the different modules of a neural network are strongly coupled, and directly transferring parameters can lead to performance degradation.
> > > We tested the tone mapper trained on Faster R-CNN by transferring it to Sparse R-CNN and Deformable DETR.
> > > The results demonstrate a  performance degradation:
> > >
> > > | Detector        | mAP         | AP50 | AP75 |
> > > | --------------- | ----------- | ---- | ---- |
> > > | Sparse R-CNN    | 34.9(-16.4) | 56.3 | 37.1 |
> > > | Deformable DETR | 39.3(-15.7) | 60.9 | 43.2 |
> > >
> > >
> > >
> > >
> > >
> > >
> > >
> > > ## Q2: The proposed performance transfer strategy can achieve modular stitching of tone mappers.
> > > In this paper, we propose a performance transfer strategy to address this issue and enable rapid performance transfer by leveraging existing pretrained weights.
> > > We use only the pretrained tone mapper (from Faster R-CNN) and detector head (from the Deformable DETR detector), performing joint fine-tuning for 5 epochs. This approach achieves high performance, comparable to fully trained results.
> > > | Detector        | map        | AP50 | AP75 |
> > > | --------------- | ---------- | ---- | ---- |
> > > | Sparse R-CNN    | 50.1(-1.2) | 75.6 | 55.7 |
> > > | Deformable DETR | 51.4(-3.6) | 76.5 | 58.7 |
> > >
> > >
> > > We implement modularity through a performance transfer strategy.

---

> > > > ### Author Response · Authors · 2025-08-07
> > > > **Looking forward to your reply**
> > > >
> > > > Dear Reviewer Vk4C:
> > > >
> > > >
> > > > Thank you for your thorough review and the valuable feedback you provided. We have tried to address all the concerns and questions you raised by providing additional experimental results, detailed clarifications, and comprehensive responses to each point you mentioned.
> > > >
> > > >
> > > > We hope these additions have adequately addressed your concerns. If you have any remaining questions or would like further clarification on any aspect of our work, we will be happy to provide additional details or experiments.
> > > > Thank you for your time and consideration.
> > > >
> > > >
> > > > Best,
> > > >
> > > >
> > > >
> > > > Authors of paper #14490

---

> > ### Author Response · Authors · 2025-08-09
> > **Rebuttal Summary**
> >
> > Dear Reviewer Vk4C
> >
> > The discussion period is ending soon. We hope this discussion has addressed your concerns regarding **joint training** and **modularity**.
> >
> > Regarding **joint training**, optimizing an image preprocessor (the tone mapper in the paper) for a specific downstream task is a well-established paradigm. It is expected to yield improved performance on the downstream task, rather than enhancing visual quality, and has been validated by extensive research in the field.
> >
> > Regarding **modularity**, while the neural network needs to be retrained, this does add complexity. To address this, we propose a performance transfer strategy that achieves strong results with minimal parameter optimizations and a limited number of training epochs. This approach also successfully preserves modularity.
> >
> > I hope these responses address your concerns and provide some insights.
> >
> > Best,
> >
> > Authors

---

### Comment · Area_Chair_e1AZ · 2025-08-01
**Author-Reviewer Discussion Period (July 31 - Aug 6)**

The author rebuttals are now posted.

To reviewers:
Please carefully read the *all* reviews and author responses, and engage in an open exchange with the authors.
Please post the response to the authors as soon as possible, so that we can have enough time for back-and-forth discussion with the authors.

---

> ### Comment · Area_Chair_e1AZ · 2025-08-05
> **Discussion Period Ends Soon (Aug 6)!**
>
> Dear reviewers,
> Thanks so much for reviewing the paper. The discussion period ends soon. To ensure enough time to discuss this with the authors, please actively engage in the discussions with them if you have not done so.

---

### Note · Authors · 2025-08-12

Dear Area Chair,

We feel it necessary to bring to your attention that some of the comments from **Reviewer[4R6p]** confuse us a lot. It seems that the comments contain factual errors.
I hope the AC will take the quality of these reviews into consideration when making their decision.


**Training Details of Comparison Methods**

**Reviewer[4R6p]**  point out that  **evaluation is unfair**,  and  **lack the details on finetuning the detector**.

We have clarified the implementation details for all comparison methods in Lines 236-243 of the paper. We cascade the tone mapper with the  detector and apply the **standard detection training strategy** (“**1x**”: COCO-pretrained weights initialization, 13 epochs, and a multi-step learning rate schedule) to **train all network parameters** in an end-to-end manner.

For the proposed **performance transfer strategy**, we perform **finetuning** only a **part of parameters** (tone mapper network and object detector) for **5 epochs**.
This is the clear meaning of **training** and **finetuning** in our paper and the other reviewers did not have any concerns regarding this.


**Misunderstanding of Experimental Results**

**Reviewer** [**4R6p**] misunderstood the term **"-NAN-"** in Table 1, interpreting it as **missing experiments** and suggesting that we did not evaluate on HDR-RAW data. However, in **Line 250** of the paper, we explicitly clarify that **"-NAN-" (Not a Number)** indicates that HDR input leads to **NAN values in the loss function, causing the network convergence to fail.**

Additionally, in the Methods section, we first mention that object detection on HDR data can cause gradient oscillation, which makes convergence difficult, and the other reviewers have no such concerns.


**Novelty**

**Reviewer** [**4R6p**] argued that our network design is simple and lacks technical novelty. **We respectfully disagree, complexity in network design does not  equate to novelty.**

Our method introduces the principle of scale invariance for tone mapping methods. Based on this principle, we employ a bias-free CNN as the tone mapper model. This simple design adheres to the scale invariance principle, avoiding unnecessary complexity.
Our experiments demonstrate that this design alone achieves state-of-the-art performance in both detection results and inference latency.
**Reviewers** [**zmDQ**] and [**2qV1**] both recognized the novelty, acknowledged the method provides some insights for HDR detection.

Best,

Authors

---

### Decision · Program_Chairs · 2025-09-17

**Decision:**

Accept (poster)

**Comment:**

This paper proposes a real-time scene-adaptive tone mapping method. The core novelty is a scale-invariant local tone mapper that ensures linear response of mapping functions to input luminance scales, enhancing detail preservation and detection performance. This method improves the accuracy and generalization capability of HDR object detection through end-to-end joint training optimization.

Reviewers acknowledge that the proposed method improves the HDR object detection and real-time performance. Meanwhile, they raise several concerns about the lack of novelty and practicality (optimization for each task).

Overall, the major weaknesses raised are:
1) Novelty and technical contribution, which falls in a scale-invariant tonemapper using scale-invariant BN.
2) There is a need for joint training of the tonemapper and each downstream task.

Despite those weaknesses, we would count the positive side of the paper; the conceptual novelty of joint training of HDR tone-mapping and downstream task (i.e., object detection) as potentially helpful for driving scenarios.